# Dynamic Optimizations of LLM Ensembles with Two-Stage Reinforcement Learning Agents

## Abstract

The advancement of LLMs and their accessibility have triggered renewed interest in multi-agent reinforcement learning as robust and adaptive frameworks for dynamically changing environments. This paper introduces RL-Focal, a two-stage RL agent framework that routes and ensembles LLMs. *First*, we develop the Decider RL-agent, which learns to dynamically select an ensemble of small size ($m_i$) among $N$ LLMs ($m_i \ll N$) for incoming queries from a user-defined downstream task $i$, by maximizing both error-diversity and reasoning-performance of the selected ensemble through iterative updates of task-adaptive rewards and policy. *Second*, to enable effective fusion of dynamically selected LLMs, we develop the stage-2 Fusion RL-agent, which learns to resolve reasoning conflicts from different LLMs and dynamically adapt to different ensemble teams composed by the Decider Agent for different downstream tasks. *Third*, we introduce the focal diversity metric to better model the error correlations among multiple LLMs further improving the generalization performance of the Decider Agent, which actively prunes the ensemble combinations. By focal diversity, we enhance performance across tasks by effectively promoting reward-aware and policy-adaptive ensemble selection and inference fusion. Extensive evaluations on five benchmarks show that RL-Focal achieves the performance improvement of 8.48% with an ensemble of small size compared to the best individual LLM in a pool and offers stronger robustness. Code is available at https://anonymous.4open.science/r/rl-focal-8DCF/

## 1 Introduction

Recently, ensemble learning has gained prominence in the domain of LLMs, with applications at the architectural level via Mixture-of-Experts (MoE) layers (Jiang et al., 2024; Liu et al., 2024a; Grattafiori et al., 2024), at the generation level through knowledge distillation and weighed combination (Wan et al., 2024; Yu et al., 2024; Huang et al., 2024; Mavromatis et al., 2024; Yao et al., 2024b), and at the output level through post-inference aggregation (Jiang et al., 2023; Tekin et al., 2024a) based on supervised learning. However, in this study, we show that ensemble learners applied to outputs of task-agnostic predictors, such as LLMs, have problem-induced instability due to the lack of adaptability, such as using an ensemble of pre-defined base learners as one solution for all. We argue that an efficient ensemble system should be adaptive and capable of (i) learning how to dynamically route diverse problems to different ensemble sets, and and (ii) learning how to fuse conflicting outputs from multiple diverse base learners, regardless of which of the ensemble sets is chosen to solve a reasoning problem.

In this paper, we present a two-stage reinforcement learning framework − RL-Focal, which aims to improve the ensemble performance by learning which ensemble strategies are most effective for a given dataset or problem by utilizing learned knowledge from previous experiments in order to adapt and learn from new situations. The design of our RL-Focal is to develop a meta-learning framework that enables a Decider RL-agent and a Fusion RL-agent to iteratively learn together from new environmental changes, e.g., the learned knowledge from previous experiments, and to adapt/update the rewards and policy for next round of learning by RL-Focal.

**Ensemble Learning in LLMs: Related Work and Open Challenges.** Ensemble training methods and the distillation (Wan et al., 2024; Yu et al., 2024; Huang et al., 2024; Mavromatis et al., 2024; Yao et al., 2024b) and the mixture of experts (MoE) (Jiang et al., 2024) methods require significant computational resources and full access to the model parameters, making it challenging to generalize to diverse contexts and adapt to domain shifts.

In post-inference aggregation ensemble learning category, most LLM ensemble research centered on supervised solutions (Jiang et al., 2023; Tekin et al., 2024a) and utilized majority voting to perform inference-time ensemble (Wang et al., 2022b; Fu et al., 2022; Li et al., 2022; Wang et al., 2022a). The downside of majority voting is the poor definition of equality between divergent answers. Two threads of research further improve majority voting, one utilizes the BLEU score as the heuristic to compare answers (Li et al., 2024), and the other enhances the BLEU score-based answer-combination method by assigning weights (Yao et al., 2024a) or creating a debate environment (Liang et al., 2023; Wan et al., 2024; Du et al., 2023; Chan et al., 2023). LLM-based Multi-Agents (Guo et al., 2024) carry similar motivations in terms of exploiting multiple LLMs to work collaboratively for a particular task. However, neither the dynamic selection of agents nor the aggregation of outputs for a given task have been systematically explored. Similarly. LLM routing aims to identify the most suitable model among the pre-defined set of LLMs for a given prompt query. Unfortunately, routing is inherently limited by the performance of the chosen model and the dependency of using another external LLM to understand/rank the matching of a prompt query to the given pool of LLMs (Ong et al., 2024).

Recently, RL approaches (Chakraborty et al., 2025; Fu et al., 2025) are proposed, which can dynamic adjust ensemble weights for an ensemble of size $N$ ($N$ is fixed for all tasks). Ensemble by RL holds the potential of creating an adaptive ensemble model (Song et al., 2023; Chua et al., 2018), ranging from time-series prediction (Liu et al., 2020; Németh & Szűcs, 2022; Perepu et al., 2020), ensemble pruning (Partalas et al., 2009; Liu & Ramamohanarao, 2020), to Tree-of-Thought (ToT) family of in-context learning of LLMs (Ouyang et al., 2022; Liu et al., 2024b; Monea et al., 2024; Zhang et al., 2021; Sun et al., 2024; Liu et al., 2024c). Yet, these existing RL approaches struggle to tackle the challenges when the ensemble of LLMs (base learners) offers very different inference performance with respect to diverse downstream reasoning tasks, given heterogeneous neural architectures fine-tuned with different LLM serving objectives as well as the challenges of whether it is feasible to dynamically compose an ensemble from a pool of base learners on demand to better address the inference performance demand of each downstream user task.

**Our contributions:** We formulate the ensemble problem as a decentralized partially observable Markov Decision Process and separate the model selection and inference fusion into two reinforcement learning stages. In Stage-1, we train a Decider RL-agent performing simultaneous actions to decide which model should be selected to serve a user-query based on the diversity metrics of the current model pool. The agent adaptively prunes the possible ensemble combinations to create the best ensemble set that minimizes the error correlation among the member models based on the focal diversity score. In Stage-2, we train the Fusion RL-agent to generate the fusion decision from different and possibly conflicting outputs generated by the member models of the selected ensemble. Extensive evaluations conducted on five benchmark datasets show that RL-Focal can surpass the best-performing base-model, outperform 12 representative SOTA LLMs tested by up to $8.48\%$, and outperforms five recent LLM ensemble approaches by up to 3% at significantly lower cost.

## 2 Preliminaries and Motivation

**Bias-Variance Trade-off and Ensemble Learning.** To demonstrate the effectiveness of EL learning bias-variance decomposition of quadratic loss is often used (Song et al., 2023). Even though the decomposition is defined for regression estimators, it is a fundamental concept that can also be generalized to any estimators, including LLMs. Assume that an estimator $\hat{f}(x)$ aims to approximate the true relation $y = f(x) + \epsilon$ by reducing the expected quadratic loss for an input $x$ and label $y$ sampled from a dataset $\mathcal{D}$:

$$\mathbb{E}[(y - \hat{f})^2] = \mathbb{E}[(\hat{f} - \mathbb{E}[\hat{f}])^2] + (y - \mathbb{E}[\hat{f}])^2 + \sigma^2 = \text{Var}(\hat{f}) + \text{Bias}(\hat{f})^2 + \text{Var}(\epsilon). \quad (1)$$

Equation 1 is the well-known bias-variance decomposition of an estimator under a given noise $\epsilon$ with zero mean and $\sigma^2$ variance (James et al., 2013). Here, $\sigma^2$ is irreducible, and as the estimator raises its complexity to approximate the true estimator, its variance will increase as it tries to capture more data points. Ensemble methods aim to reduce the bias and variance jointly e.g., by representing the parts of the hypothesis space with each estimator (Dietterich, 2000). Following (Krogh & Vedelsby, 1994), one can present the *ambiguity decomposition* by defining the ensemble model as the convex combination of its component models: $\hat{f}_{\text{ens}} = \sum_i w_i \hat{f}_i$ where $\sum_i w_i = 1$. The ambiguity decomposition shows that the quadratic error of the ensemble estimator is guaranteed to be less than or equal to the average quadratic estimators of its component estimators, which is formalized as follows:

$$(y - \hat{f}_{\text{ens}})^2 = \sum_i w_i (\hat{f}_i - y)^2 - \sum_i w_i (\hat{f}_i - \hat{f}_{\text{ens}}). \quad (2)$$

Here, the first term is the weighted average error of individual estimators and the second term is the *ambiguity term* showing the variance between the individual estimators. Thus, the second result of this decomposition is that the greater the ambiguity, i.e., the higher error correlation between individual estimators, the lower overall error the ensemble may result. More explicitly, according to (Brown et al., 2005), one can substitute the ensemble estimator, $\hat{f}_{\text{ens}} = \frac{1}{M}\sum_i \hat{f}_i$ in Equation 1 to break down the variance component even further to obtain *bias-variance-covariance* decomposition:

$$\mathbb{E}[(\hat{f}_{\text{ens}} - y)^2] = \overline{\text{Bias}} + \frac{1}{N}\overline{\text{Var}} + (1 - \frac{1}{N})\overline{\text{Covar}}. \tag{3}$$

As the averaged covariance term implies, the quadratic loss of ensemble networks depends on the error correlation among its estimators. Thus, the selected estimators used to construct an ensemble learner are expected to make uncorrelated errors in order for the ensemble to obtain a lower overall error, and each of the component estimators should cover a part of the hypothesis space to ensure that the average bias and variance are lower.

**Weighted Consensus of Ensemble Learner and its Instability.** Modern deep learning models, incl. LLMs, target cross-domain generalization. The few-shot learners are the pioneering efforts for this capability. The $k$-shot models are able to learn the relation between the input and the label for a task $\mathcal{T}$ with a very small number of samples, i.e., $(x_i, y_i) \sim \mathcal{T}$ for $1 \leq i \leq k$ where $k$ in the range of $1, \ldots, 5$. The zero-shot models learn to produce the desired output with no $y$ given. Let $w_{\text{best}}$ denote the best weight assigned to each estimator for a given task $\mathcal{T}$. An ensemble estimator $\hat{f}_{\text{ens}}$, created by the convex combination of its component models, is fit to the task $\mathcal{T}$ as follows:

$$w_{\text{best}} = \arg\min_w \mathbb{E}_{(x,y)\sim\mathcal{T}}[(\sum_i w_i \hat{f}_i(x) - y)^2]. \tag{4}$$

The problem with such an ensemble estimator arises when the task changes over time or when the contexts are different. The current weights, $w$, are fitted for the given task $\mathcal{T}$ under a given context or at a given time. But when the task changes, the weights representing the importance of each estimator may lose their cross-estimator assessment validity and create instability. Consider a pool of $N$ LLMs, one LLM can be good at commonsense-reasoning, while another is good at STEM-related topics, and so forth. In this paper we argue that an ideal way of composing an ensemble ensemble estimator $\hat{f}_{ens}$ from the pool of candidate LLMs is **task-adaptive**, i.e., to construct the ensemble that is most-effective for each given task by selecting a subset of models in the pool, which offers the task-specific strength and complimentary wisdom. Our experiments have shown that an ensemble estimator of smaller size with high error diversity can outperform the large ensemble of all LLMs with better generalization performance and at lower runtime cost.

## 3 RL-FOCAL: DESIGN METHODOLOGY

We first give an architectural overview in **Figure 1** with highlight of five steps: (1) We examine the current model pool $\mathcal{E}_t$, assuming the pool initially has $N$ LLMs, and the diversity metrics are calculated to create the observation of the Decider Agent (see Section 3.4 for detail). (2) The Decider RL-Agent learns to select models to create a new pool denoted as $\mathcal{E}_{t+1}$ of size $m$ LLMs for a task-specific query $x_t$ where $m \ll N$. (3) The input query $x_t$ is sent to each of the LLMs in the new pool to obtain the Fusion Agent's observation. (4) The Fusion Agent learns to combine their outputs to make the fusion decision. (5) Both the Decider RL-Agent and Fusion RL-Agent are trained with the global state and fusion results via reinforcement learning using a Centralized Critic Net (Schulman et al., 2017). This process iterates until all episodes are finished, or it continues indefinitely in an online setup.

**Problem Formulation.** Let $\mathbf{x}_1, \ldots, \mathbf{x}_n$ denote a sequence of input queries/prompts with length $n$, where each prompt $\mathbf{x}_i$ is sampled from a task $\mathcal{T}_j$, denoted by $\mathbf{x}_i \sim \mathcal{T}_j$. The queries can originate from a single task or from a group of tasks $\mathcal{T}_1, \mathcal{T}_2, \ldots, \mathcal{T}_J$, e.g., Math, Biology, and History with varying difficulties. Consider a query $\mathbf{x}$ sampled from these tasks is targeted to an ensemble of $N$ LLMs, denoted by $\mathcal{E}_t = \{M_1, \ldots, M_N\}$ to obtain the desired output $\mathbf{y}$ at time $t$. As the task distribution and difficulty levels of the queries alter over time, a non-stationary environment is formed to which this pool of $N$ LLMs needs to adapt. We assume a regular temporal cycle, such as a Math-related query likely followed by another Math query, and task switches are possible but not in high frequency. In this setting, the **first** problem is to dynamically find a small subset of $m$ LLMs from the pool that are the most suitable for the query task, and learn to infer the best ensemble team of the selected $m$ models, and generate the set of outputs $\hat{\mathbf{y}}_1, \ldots, \hat{\mathbf{y}}_m$, where $1 \leq m \leq N$. This motivates the design of our Decider RL-Agent. The **second** problem is to resolve the potential reasoning conflict among

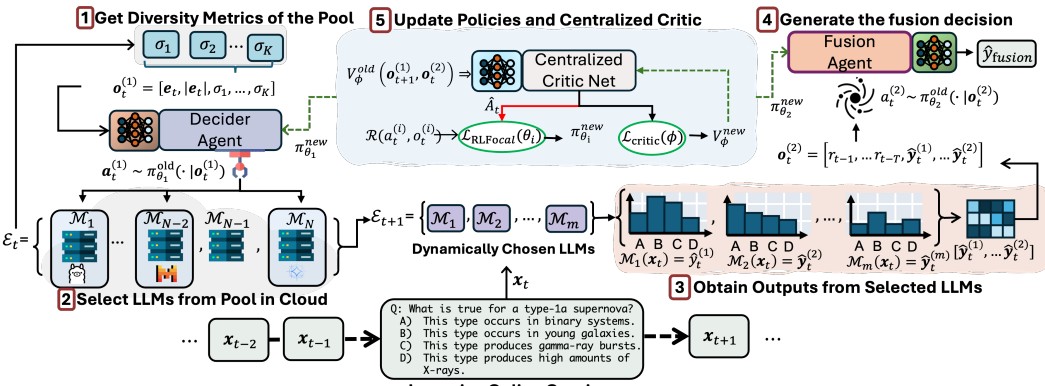

Figure 1: Overview of *RL-Focal* two stage ensemble by reinforcement learning agents.

the generated outputs from the selected ensemble of $m$ LLMs to make the final ensemble decision, $\hat{y}_{\text{fusion}}$, such that the ensemble error, i.e., $\hat{y}_{\text{fusion}} - y$, is minimized. This motivates the design of Fusion RL-Agent. Each stage exhibits temporal dependence and dynamics, requiring an exploitative approach to identify the best possible model selection and provide a fusion-enhanced solution using feedback from the environment in the form of task-adaptive rewards and decision policy.

Based on the objectives of the two-stage solution and the dynamics of the environment, we study a decentralized partially observable Markov decision process (DEC-POMDP) with shared rewards. For each objective, we define an agent: the first is the *Decider Agent*, and the second is the *Fusion Agent*. The agents are fully cooperative in minimizing $\hat{y}_{\text{fusion}} - y$ and acting independently based on local observations. Further, the second agent's observation depends on the actions of the first agent, and thus it is an extensive-form game (Zhang et al., 2021). A DEC-POMDP has the elements $(\mathcal{S}, \mathcal{A}, \mathcal{P}, \mathcal{O}, \mathcal{R}, n, \gamma)$ (Yu et al., 2022) and, in our context, we define them as follows: $\mathcal{S}$ is the state space, with $\mathbf{s} \in \mathcal{S}$. $\mathbf{o}^{(i)} = \mathcal{O}^{(i)}(s)$ is the local observation of agent $i$ creating the state vector $\mathbf{s} = [\mathbf{o}^{(1)}, \mathbf{o}^{(2)}]$. $n$ is the number of agents and $n = 2$ in our context. The joint action space is denoted by $\mathcal{A}$ and $P(\mathbf{s}'|\mathbf{s}, \mathbf{a})$ is the transition probability from $\mathbf{s}$ to $\mathbf{s}'$ given by actions of agents $\mathbf{a} = [\mathbf{a}^{(1)}, \mathbf{a}^{(2)}] \in \mathcal{A}$. The shared reward is represented by $\mathcal{R}(\mathbf{s}, \mathbf{a}, \mathbf{s}')$ and $\gamma$ is the discount factor. Each agent uses its policy $\pi_{\theta_i}(\mathbf{a}^i|\mathbf{o}^i)$ parametrized by $\theta_i$ to produce an action $\mathbf{a}^i$ based on its local observations and jointly optimize the accumulated discounted reward: $\mathbb{E}_\pi \left[ \sum_{t>0} \gamma^t r_t \right]$ where $r_t$ denotes reward at time step $t$ and $\pi$ is the joint policy. We obtain the best parameters of the policy functions by following Multi Agent Proximal Policy Optimization (MAPPO) (Yu et al., 2022) with the Centralized Critic Net approximating the value of the current state (see Section G for more detail).

### 3.1 ACTIVE ENSEMBLE PRUNING WITH DECIDER AGENT

The Decider RL-agent is responsible for selecting the "best" models for the incoming query to minimize redundant or unnecessary inferences. The ensemble selection should be performed by respecting the error correlation among the member models of each ensemble in order to choose the best ensemble that can effectively lower the squared error (recall Section 2). Multiple diversity metrics (see Section 3.4) are used to learn the selection of the best ensemble team among a total of $2^N - N - 1$ ensemble teams of size ranging 2 to $N$, given a pool of $N$ LLMs. Unlike the conventional diversity-based ensemble approaches Brown et al. (2005); Tekin et al. (2024a), which examine all possible combinations with diversity scores and accuracy measures to make the ranking decision, we introduce RL based approach to navigate in the surface of diversity and accuracy. The Decider RL-agent observes the diversity of the current model pool while aiming for the accuracy boost. It periodically updates its policy to adapt to the changes in the environment, so that it can actively add or remove models from the current pool. We define the elements of a Decider Agent as follows:

**State:** For $K$ number of diversity metrics denoted by $\sigma_1, \ldots, \sigma_K$, the agent observes $\mathbf{o}_t^{(1)} = [\mathbf{e}_t, \|\mathbf{e}_t\|_1, \sigma_1, \ldots, \sigma_K]$ at time $t$, where $\mathbf{e}_t = (e_1, \ldots, e_N), e_i \in \{0, 1\}$ is the binary vector representing the current model pool where $e_i = 1 \iff M_i \in \mathcal{E}_t$ and $\|\mathbf{e}_t\|_1$ is the current size of the pool. The diversity metrics are calculated based on the historical data with a window size $T$. The details of the designed metrics are given in Section 3.4. **Action:** The agent simultaneously decides whether each model should be included in the model pool. Accordingly, we define the action at time

$t$ as a binary vector $\mathbf{a}_t^{(1)} \in (a_1, \ldots, a_N), a_i \in \{0, 1\}$, where each $a_i$ indicates whether the model with index $i$ is included in the pool. Each action $a_i$ is independent of the others, i.e., the selection of one model does not depend on the selection of the others. **Policy:** At time $t$, the policy provides a probability vector $\pi_{\theta_1}(\mathbf{a}_t^{(1)} \mid \mathbf{o}_t^{(1)}) = [p_1, p_2, \ldots, p_N]$. $p_i = \Pr(a_i = 1 \mid \mathbf{o}_t^{(1)}; \theta_1)$ is the probability for model $i$ to be included in the model pool, $\theta_1$ is the policy parameters of Decider agent, and the action $a_i$ is drawn from Bernoulli distribution with success probability $p_i$, i.e., $a_i \sim \text{Bernoulli}(p_i)$.

The Decider agent makes multiple independent decisions simultaneously, which can be modeled in RL at multi-action settings. We employ a branching solution to model each action branch with another parameter set (Tavakoli et al., 2018). Specifically, the Decider Agent's policy parameterized by a Multi-layer Perceptron (MLP) consists of fully connected layers with sigmoid activation functions. The final layer branches into separate heads for each action with its own set of weights:

$$
\begin{aligned}
\mathbf{z} &= \rho(\mathbf{W}_{L-1}(\ldots \rho(\mathbf{W}_1 \mathbf{o}_t^{(1)}) \ldots)), \\
p_1 &= \rho(\mathbf{W}_L^{(1)} \mathbf{z}), \ldots, p_N = \rho(\mathbf{W}_L^{(N)} \mathbf{z}),
\end{aligned}
\tag{5}
$$

where $\mathbf{W}_j$ is the weight matrix at layer $j$, $\rho$ represents the sigmoid activation, $z$ is the penultimate layer outputs, and $L$ is total number of layers where $j = 1, \ldots, L$. The initial layers extract from the current observation vector, $\mathbf{o}_t^{(1)}$, while the final parameters, $\mathbf{W}_L^{(1)}, \ldots, \mathbf{W}_L^{(N)}$, independently model the probability of each model being included in the next model pool. Therefore, during training, the first layers are jointly trained while the last layers are tuned for each model separately. **Transition:** The observation vector contains stochastic term which govern by joint distribution of independent Bernoulli trials, $P(\mathbf{e}_{t+1} | \mathbf{o}_t^{(1)}, \mathbf{a}_t^{(1)}) = \Pi_i^N p_i^{e_i}(1-p_i)^{1-e_i}$, and also deterministic terms $\left[ \|\mathbf{e}_{t+1}\|_1, \sigma^{(1)}, \ldots, \sigma^{(N)} \right]$. **Reward:** The shared reward is the most important metric in our design since it defines the objective of making $\hat{y}_{\text{fusion}} = y$ for both RL-agents. To this end, we define the reward function $\mathcal{R} : (\mathbf{a}_t, \mathbf{o}_t, y) \to r_t$, mapping the agent observation and action at time $t$ to reward value $r_t$:

$$
\mathcal{R}(\mathbf{a}_t, \mathbf{o}_t, y) = \begin{cases} 1 & \text{if } \hat{y}_{\text{fusion}} = y, \\ -1 - \alpha \cdot \frac{\|\mathcal{E}_t\|_1}{N} & \text{otherwise}, \end{cases}
\tag{6}
$$

where $\alpha \in [0, 1]$ is the size-penalization constant to force the Decider Agent to decrease pool size.

**Remarks:**(1) The reward requires the final decision at time $t$, $\hat{y}_{\text{fusion}}$, generated by the Fusion Agent and the correct output $y$, as illustrated in the steps of Figure 1. (2) Multi-agent RL systems carry stability issues due to agents exhibiting mutual dependence with limited observations. We observed that performing a warm start resulted in more stable training. However, to perform a warm start on the Decider Agent, we need an evaluator metric to evaluate the created model pool. Thus, we substitute $\hat{y}_{\text{fusion}}$ with an interim prediction using plurality voting, which chooses the most voted decision based on the current model pool. The interim prediction stabilized the training and helped the Decider Agent's policy network to converge. We provide an offline warm-start training procedure in Algorithm 1 and discuss the details in Appendix G.

### 3.2 Generating Final Decision with Fusion Agent

Fusion Agent is responsible for reaching the final decision based on the possibly conflicting outputs generated by the member models in the current pool, which forms the selected ensemble at the current iteration $t$. The success of the Decider agent would be undervalued if the Fusion agent fails to resolve the disagreement of the outputs to reach the correct final decision. As demonstrated in (Dietterich, 2000), ensemble models can computationally achieve the global optimum by leveraging the local optima of individual models as starting points. We advocate that the generated outputs by each model (e.g., the probabilities assigned to each option in a multiple-choice question (MCQ)) may indicate/locate the vicinity of the global optimum, and the Fusion agent can perform a convex combination to reach the optimum. We define the elements of the Fusion Agent as follows:

**State:** The agent observes the outputs generated by the models in the $m$-sized pool and past interactions with the environment. The output of models is a sequence of words for open-ended questions (OEQ). In the case of MCQ, we can represent the observation as the probabilities assigned to each choice (see Appendix F for more details). Then, the observation of the Fusion Agent is $\mathbf{o}_t^{(2)} = [r_{t-1}, \ldots, r_{t-T}, \mathbf{p}_1, \ldots, \mathbf{p}_m]$, where $\mathbf{p}_i = (p_{i1}, \ldots, p_{ik}) = M_i(\mathbf{x}_t)$ is the probability vector that model $M_i$ assigned to the choices for input $\mathbf{x}_t$ denoted by $p_{ij} = \Pr(\hat{y}_j \mid \mathbf{x}_t; \psi_{M_i})$ and $k$ is the number of choices and $\psi_{M_i}$ is model $i$ parameters (which remain frozen and inaccessible). Here, $r_{t-1}, \ldots, r_{t-T}$ represents the past rewards until time $t-T$. **Action:** Based on the current observation,

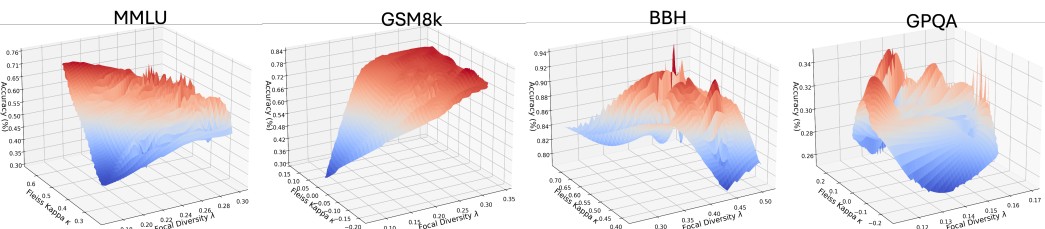

Figure 2: All candidate ensemble teams from the model pool are plotted with their focal diversity scores, Fleiss Kappa, and Accuracy using the 4 popular LLM evaluation datasets. We use cubic interpolation to create a surface, and the dark red represents a higher performance score.

the agent makes the final decision $a_t = \hat{y}_{\text{fusion}}$, which we define as the action that the agent can take. In the case of multiple-choice questions, we can define the action at time $t$ as $a_t \in \mathcal{A} = \{0, \ldots, k\}$, where each index indicates a choice and $\mathcal{A}$ is the action space. **Policy:** The policy of Fusion Agent produces a probability distribution with the size of choices, and the action is the choice that maximum probability assigned, i.e. $a_t = \arg\max_{a \in \mathcal{A}} \pi_{\theta_2}(a \mid \mathbf{o}_t^{(2)})$. Specifically, we parameterize the policy with an MLP containing multiple layers of fully connected weights and sigmoid activation functions as a Fusion policy network. Here we focused on the MCQ, yet in Appendix H we show that Decider Agent can be extended to OEQs. We recommend referring to the studies Jiang et al. (2023); Tekin et al. (2024a;b) as foundational resources for developing an ensemble policy network for OEQ. **Reward:** We use the same reward equation presented in Equation 6 for our Fusion agent, excluding the size-penalization constant. Similarly, we initialize the model parameters with a warm start where we use the outputs from the warm-started Decider Agent. **Transition:** The transition of this agent is deterministic which is defined by $\mathbf{o}_{t+1}^{(2)} = [r_t, \ldots, r_{t-T}, \mathbf{p}_1, \ldots, \mathbf{p}_m]$, where $\mathbf{p}_i = M_i(\mathbf{x}_{t+1})$ is the model output for input $\mathbf{x}_{t+1}$.

### 3.3 UPDATE RULE BY RL-FOCAL ALGORITHM AND CENTRALIZED CRITIC NETWORK

**Figure 2** is the visual evidence of how the performance of different model combinations evolves as the task associated with incoming queries changes. The role of the RL-Focal is to walk on the surface created by the diversity metrics and explore a model combination that gives high performance. Unlike the previous works (Tekin et al., 2025; 2024a), which perform exploration offline on a supervised dataset with the Genetic Algorithm, RL-Focal makes the exploration online by actively forming the ensemble on the downstream task using RL. Such online exploration enables the RL-focal agent to timely adapt to a changing environment, e.g., evolving query tasks, changing policy for selection and fusion of relevant models.

Concretely, the policies are updated with the new parameters by loss functions $L_{\text{RLFocal}}$ and $L_{\text{Critic}}$. The reward trajectory $\tau$ is formed as an agent (Decider/Fusion) iteratively collects reward by executing the current policies on input queries, followed by policy and value function optimization, to achieve the highest possible discounted cumulative reward $J(\theta) = \mathbb{E}_{\tau \sim \pi_\theta}[R(\tau)]$. The policy network parameters should be optimized to increase the probability of action-state pairs that yield positive rewards. To achieve this, we can perform gradient ascent optimization per agent by calculating $\nabla_\theta J(\theta) = \sum_t \nabla_\theta \log \pi_\theta(a_t \mid s_t) R(\tau)$. Nevertheless, RL-Focal is a multi-agent system with two RL-agents, where each RL-agent has its own observation space, which makes the environment non-stationary. To address this and stabilize optimization, we employ the following loss function by leveraging MAPPO:

$$\mathcal{L}_{\text{RLFocal}}(\theta) = \frac{1}{n} \sum_{k=0}^{n} \min\left(\hat{r}_t^{(k)}(\theta_i)\hat{A}_t, \text{clip}(\hat{r}_t^{(k)}(\theta_i), 1 - \epsilon, 1 + \epsilon)\hat{A}_t\right), \quad \hat{r}_t(\theta_i) = \frac{\pi_{\theta_i^{\text{new}}}(a_t|\mathbf{o}_t)}{\pi_{\theta_i^{\text{old}}}(a_t|\mathbf{o}_t)} \quad (7)$$

where $\hat{r}_t$ is the ratio term for each agent, $\hat{A}_t$ is the estimated advantage function, and is common for both agents. The clip function ensures that the policy updates are stable by keeping the ratio terms within the range $[1 - \epsilon, 1 + \epsilon]$. The advantage function measures how much better the joint action is compared to the average performance of policies in the global state and it is estimated using (GAE) (Schulman et al., 2018): $\hat{A}_t(\mathbf{s}_t, \mathbf{a}_t) = \sum_{l=0}^{\infty} (\gamma\lambda)^l \delta_{t+l}$ by calculating $\delta_t = r_t + \gamma V_\phi(s_{t+1}) - V_\phi(s_t)$ where $s_t = [\mathbf{o}_{t+1}^{(1)}, \mathbf{o}_t^{(2)}]$ is the global state which feed into Critic Network $V_\phi$ to estimate the expected return if we follow the policy from state $\mathbf{s}_t$. The central Critic observes the newly created pool diversity and its outputs to estimate the value (see more details in Appendix G).

Overall, the central Critic creates a bridge between two agents with global information and reduces the non-stationarity to stabilize the training. We optimize the Critic's parameters by calculating the MSE between value predictions and target values: $\mathcal{L}_{\text{Critic}}(\phi) = \frac{1}{2}\mathbb{E}_t\big[(V_\phi(\mathbf{s}_t) - \hat{V}_t)^2\big]$ where $V_\phi(\mathbf{s}_t)$ is the value prediction for state $\mathbf{s}_t$ and $\hat{V}_t$ is the target value computed by the reward $\hat{V}_t = \sum_{l=0}^{T-t}\gamma^l r_{t+l}$. Building on these formulations, we first initialize the parameters of the Agents and Critic by employing the offline warm-start Algorithm 1. During the online execution, the agents are then periodically updated with the online Algorithm 2 to ensure stability and adaptability to the incoming queries.

### 3.4 Diversity Metrics and Focal Diversity

For a pool of $N$ base models, the total number of possible ensemble teams with size $m$ is $2^N - N - 1$, where $2 \leq m \leq N$. To reduce the overhead of considering all possible combinations, the decider agent enables effective ensemble pruning by adaptively selecting ensembles with high error diversity (aka low error correlation) by utilizing two new metrics, specifically designed to capture error correlation among the member models of an ensemble.

**Focal Negative Correlation & Focal Diversity.** The focal negative correlation metric $\rho^{focal}$ is used to quantify the level of error diversity among the component models of an ensemble concerning each model within the ensemble. The focal diversity metric $\lambda^{focal}$ is used to quantify the general error diversity of the ensemble by taking into account all $\rho^{focal}$ in the ensemble. We choose one of the $N$ base models each time as the focal model to compute the focal negative correlation score of this ensemble, denoted as $\rho^{focal}(\mathcal{M}_i; \mathcal{E})$. We define the focal diversity of this ensemble team by the average of the $N$ focal negative correlation scores. The procedure of computing the focal negative correlation score of $\rho^{focal}$ is as follows: (i) select a model among the set of $N$ models as the *focal* model, (ii) extract all queries from the historical data within a time window of length $T$ where the focal model has failed, and compute the focal negative correlation score (iii) repeat the previous steps until all $N$ focal negative correlation scores are obtained. $\rho_1^{focal}, \ldots, \rho_N^{focal}$, and (iv) compute the average over the scores to obtain the focal diversity of ensemble $\mathcal{E}$, denoted by $\lambda^{focal}(\mathcal{E})$:

$$\lambda^{focal}(\mathcal{E}) = \frac{1}{N}\sum_{\mathcal{M}_i \in \mathcal{E}}\rho^{focal}(\mathcal{M}_i; \mathcal{E}), \ \rho^{focal}(\mathcal{M}_i; \mathcal{E}) = 1 - \frac{\Pr(\mathcal{K}=2)}{\Pr(\mathcal{K}=1)} \tag{8}$$

The term $\mathcal{K}$ is a random variable that represents number of models simultaneously failing on an test input, e.g., $\Pr(\mathcal{K}=2)$ represents the probability of two randomly chosen models simultaneously failing on an input. We calculate $\Pr(\mathcal{K}=2) = \sum_{j=1}^{N}\frac{j(j-1)}{N(N-1)}p_j$, $\Pr(\mathcal{K}=1) = \sum_{j=1}^{N}\frac{j}{N}p_j$ and $p_j$ is the probability that $j$ models fail together on a randomly chosen input. It is measured by $p_j = n_j/T$ where $n_j$ is the total number of inputs that $j$ models failed together on a set of test inputs and $T$ is the total number of queries. The terms beneath $p_j$ values, e.g. $\frac{j(j-1)}{N(N-1)}$, are the probability of the chosen model being one of the failure modes. In the case of minimum diversity, the probability of two randomly chosen models failing together comes down to the probability of one of them failing, which makes the fraction term equal to 1 and $\rho^{focal} = 0$. Similarly, in the case of maximum diversity, there are no simultaneous failures. Hence, the nominator equals 0 and $\rho^{focal} = 1$. **Figure 2** shows that compared to the common metrics e.g., Fleiss' Kappa (Fleiss & Cohen, 1973), which measures the amount of agreement, the focal diversity is highly correlated with the generalization performance of an ensemble across all four benchmark datasets: MMLU, GSM8K, BBH, and GPQA. A theoretical proof for the robustness of Focal Diversity is given in Appendix K.

| Model Name | GSM8k (Acc %)↑ | MMLU (Acc %)↑ | GPQA (Acc %)↑ |
|---|---|---|---|
| Llama-3.1-8B-Instruct | 84.5 | 66.8 | 32.8 |
| Qwen-2-7B-Instruct | 85.7 | 65.3 | 34.3 |
| Qwen-2.5-7B-Instruct | 91.6 | 68.2 | 36.4 |
| PAIRRANKER Jiang et al. (2023) | 91.7 | 63.8 | 34.2 |
| GAC Yu et al. (2024) | 88.1 | 67.5 | 33.4 |
| DEEPEN Huang et al. (2024) | 86.2 | 67.1 | 32.6 |
| RLAE-PPO Fu et al. (2025) | 91.3 | 68.5 | 36.1 |
| RLAE-MAPPO Fu et al. (2025) | 92.5 | 69.1 | 35.3 |
| RL-Focal | **93.26** | **69.2** | **36.78** |

Table 2: RL-Focal performance compared to RLAE and the other Ensemble methods.

| Methods | MMLU | | GPQA | |
|---|---|---|---|---|
| | Accuracy (%)↑ | Cost (¢)↓ | Accuracy (%)↑ | Cost (¢)↓ |
| Llama-3.1-8B-Instruct | 66.8 | 2,808 | 32.8 | 215.5 |
| Qwen-2-7B-Instruct | 65.3 | 16,848 | 34.3 | 1,293 |
| Qwen-2.5-7B-Instruct | 68.2 | 3,931 | 36.4 | 301.7 |
| RLAE-MAPPO | 69.1 | 19,656 | 35.3 | 1,810.2 |
| RL-Focal | **69.2** | **8,486** | **36.78** | **274.1** |

Table 3: Performance and cost on MMLU and GPQA for RL-Focal and RLAE-MAPPO Fu et al. (2025). We use the same model pool utilized by authors of Fu et al. (2025) and create our dynamic ensemble pool.

### 4 Experimental Evaluations

**Performance of RL-Focal.** The first set of experiments contains 4 different benchmarks in MCQ format: MMLU (Hendrycks et al., 2020), BBH (Suzgun et al., 2022), MUSR (Sprague et al., 2023),

| Model Name | MMLU (Acc %)↑ | GSM8k (Acc %)↑ | BBH (Acc %)↑ | MUSR (Acc %)↑ | GPQA (Acc %)↑ |
|---|---|---|---|---|---|
| Phi-2b | 55.82 | 68.85 | 44.55 | 41.90 | 28.89 |
| Gemma-2b | 40.26 | 24.03 | 11.76 | 1.68 | 11.43 |
| Gemma-7b | 63.87 | 73.04 | 36.23 | 46.59 | 27.78 |
| Llama-2-7b | 41.79 | 10.87 | 10.35 | 3.76 | 2.24 |
| Mistral-7b | 59.67 | 56.21 | 22.17 | 10.68 | 5.59 |
| Llama-2-13b | 53.40 | 41.74 | 39.66 | 44.90 | 28.89 |
| Phi-4-14b | 80.7* | 92.2* | 59.94 | 42.23 | 32.22 |
| Gemma-2-27b | 75.2* | 74.0* | 47.74 | 46.69 | 32.22 |
| Llama-2-70b | 68.53 | 58.89 | 28.03 | 41.54 | 30.00 |
| Mixtral-8x7b | 70.42 | 73.91 | 41.87 | 48.85 | 31.11 |
| Mixtral-8x22b | 76.36 | 88.2* | 53.94 | 48.03 | 28.89 |
| Qwen-2.5-72b | 82.68† | 91.99† | 57.53 | 51.97 | 45.56 |
| Llama-3-70b | 77.29 | 84.5* | 54.88 | 53.95 | 40.00 |
| Deepseek-LLM-67b | 71.24 | 63.4* | 44.90 | 51.31 | 38.89 |
| RL-Focal | **83.59** ± 0.73 | **93.89** ± 0.74 | **65.00** ± 1.21 | **55.25** ± 0.32 | **48.28** ± 0.59 |
| Rel. Gain | +1.10 | +2.07 | +8.48 | +2.40 | +5.97 |

Table 1: RL-Focal selection and ensemble composition for the evaluated LLM benchmarks. Models marked with * denote scores sourced from their respective technical reports and are therefore excluded from the candidate ensemble pool. For each dataset column, any model score not marked with * is included in the candidate model pool, and we obtain its score. † Turbo version of the model is used

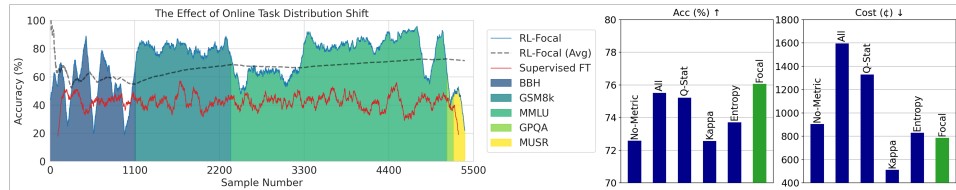

Figure 3: The left shows performance for RL-Focal for online dataset distribution, and we compare with the SFT Ensemble MLP that is trained using BBH and tested under data shift. The last two plots show how diversity metrics affect the performance and cost of the RL system on the GSM8k dataset.

and GPQA (Rein et al., 2023) are the benchmarks present in the HuggingFace leaderboard (Beeching et al., 2023). We also add GSM8K (Cobbe et al., 2021), which contains open-ended math problems.

The main results for the 5 benchmarks are shown in **Table** 1. The LLM pool contains 14 open-source LLMs ranging from 2b to 70b parameters. We make two observations. (1) RL-Focal outperforms the best LLM performance for all 5 benchmarks, i.e., Mixtral-8×22b (MMLU), Mixtral-8×7b (GSM8k), Phi-4-14b (BBH), Llama-3-70b (MUSR), and Qwen-2.5-72b (GPQA). (2) Specifically, RL-Focal outperforms Mixtral-8×7b, an ensemble by training with MoE, by 7.56% on MMLU, 4.93% on GSM8k, 23.13% on BBH, 6.40% on MUSR, and 17.17% on GPQA datasets. The results indicate that the Decider Agent can effectively select the best ensemble set for each query task on demand by updating the base model pool based on focal diversity scores of different ensemble sets and the task-aware rewards and policy learned. The Fusion Agent can effectively exploit the disagreements among the component models of the selected ensemble to generate high-quality final output. Error bars are shown only for RL-Focal, as the base model uses a fixed inference set and therefore exhibits no variability. We report the mean and standard error over 5 runs of MARL-Focal, all conducted with the same base model inference and provide the reward curves in Appendix C.

The **Table** 2 compares RL-Focal with RLAE and other ensemble methods in the literature by utilizing the same model pool. While other methods struggle to improve base model performance, RL-Focal improves the performance by 1.1% in MMLU and 0.38% in GPQA. RLAE-MAPPO shows comparable performance, yet **Table** 3 shows that RL-Focal reduces the cost by factors of 2× and 9×. The first plot in **Figure** 3 shows the advantage of RL-Focal's RL nature over a supervised-tuned fusion model under data distribution shift. We started from the BBH dataset and let Algorithm-2 fit the incoming data distribution by performing model selection and fusion (see Appendix B for dynamic change of model-pool and Appendix D for more details on fusion model). The SFT method is tuned on the BBH training set and demonstrates moderate performance on BBH, but cannot fit the other sets and makes redundant selections. On the other hand, bar-charts in Figure 3 show the impact of Focal Diversity on RL-Focal. In the *No-metric* setup, the Decider Agent selects models solely based on environmental rewards. In the *All* setup, model selection is bypassed and the entire model pool is used. We also compare Focal Diversity with three existing popular diversity metrics. Overall, Focal Diversity achieves the highest accuracy and second-lowest cost, while Kappa diversity incurs the lowest cost with the worst accuracy.

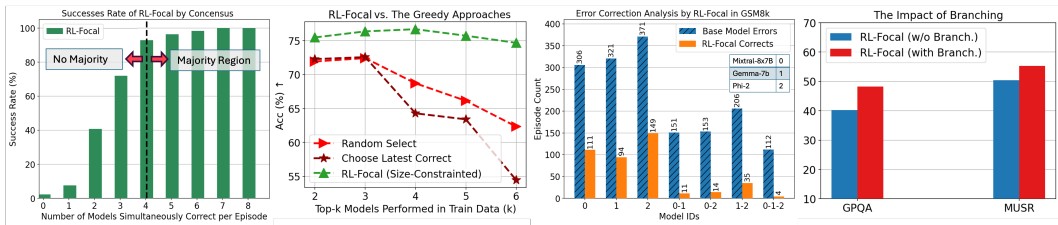

Figure 4: The first plot shows how often RL-Focal is correct when exactly $n$ base models are correct (x-axis). The plot in the middle shows the performance of RL-Focal compared to two greedy approaches. The third plot shows how often RL-Focal corrects simultaneous errors made by top-performing base models. The last plot shows the improvement by the branching of Decider Agent.

| Method | 1st Model | 2nd Model | GPQA |
|---|---|---|---|
| RouteLLM | Llama-3-70b | Mixtral-8x7b | 40.00 |
| RouteLLM | Llama-3-70b | Gemma-2-27b | 38.88 |
| RouteLLM | Llama-3-70b | Qwen2.5-72b | 36.66 |
| RL-Focal | - | - | **48.28** |

Table 4: Comparison of RL-Focal to RouteLLM using three combinations of strong models.

| Method | Model ID | MMLU | GSM8k |
|---|---|---|---|
| More Agents (Li et al., 2024) | 6 $\lceil\times 40\rceil$ | 51.09 | 61.00 |
| More Agents (Li et al., 2024) | 7 $\lceil\times 40\rceil$ | 60.05 | 77.00 |
| LLM-Blender (Jiang et al., 2023) | 12345678 | 44.01 | 40.41 |
| Majority Voting | 12345678 | 68.06 | 72.31 |
| Mixtral-8x7b | 8 | 70.53 | 71.16 |
| DyLAN (Liu et al., 2024c) | - | 70.5 | - |
| LLM-TOPLA (Tekin et al., 2024a) | 378 $\mid$ 138 | 72.77 | **79.01** |
| RL-Focal | Dynamic | **77.98** | 78.84 |

Table 5: Comparison with 6 other ensemble methods.

**Table 4** reports the performance comparison of RL-Focal with RouteLLM (Ong et al., 2024) on GPQA, showing RL-Focal outperforms 3 combinations of strong models in RouteLLM by a large margin of $8.28\% - 11.62\%$. **Table 5** shows the comparison of RL-Focal with 6 existing representative ensemble methods on MMLU and GSM8k. We make two observations. (1) RL-Focal shows the best performance on MMLU with an overall improvement of $5.21\% - 33.97\%$ improvement. (2) RL-Focal offers on par performance on GSM8k to LLM-TOPLA, a supervised approach, at significantly lower cost (See cost analysis in Appendix I) but effectively outperforms More Agents with 40 LLMs on MMLU by 17.93% and TOPLA by 5.21% with the initial pool of only 12 LLMs as listed in Table 1.

**Ablation Study of RL-Focal. Figure 4** reports the results of experiments on the effect of consensus on RL-Focal with four plots. The first plot shows that the success rate is nearly maximal in the Majority region and remains as high as $40\%$ even when only 2 out of 8 models are correct. The second plot shows a comparison of RL-Focal with two greedy approaches: (i) Random Select: selects the top-$k$ models based on training performance and randomly picks one of their outputs at test time. (ii) Latest correct: Selects the output of the most recently model which is correct from the previous step within a pool of $k$ models; if multiple models were correct, one is chosen at random. As shown in the plot, RL-Focal is better and stays effective as the number of models in the pool increases, while the greedy approaches suffer significantly. The third plot illustrates how RL-Focal corrects errors made by the top-3 best-performing individual models and their combinations on the GSM8k benchmark. Even when the majority of these models made incorrect decisions, RL-Focal effectively generates the correct one thanks to the dynamic ensemble enabled by its two-stage RL-agent framework (see Appendix J). The last plot in Figure 4 shows that the Decider Agent with the branching for each agent at the final layer, compared to a single branch, improves the performance by $8.05\%$ and $4.79\%$ in GPQA and MUSR datasets.

## 5 CONCLUSION

We presented RL-Focal, a novel two-stage RL agent approach to ensemble learning of LLMs with three original contributions. First, we formulate the ensemble problem as a DEC-POMDP and separate the model selection and inference fusion into two RL stages. Second, in Stage-1, we train a Decider RL-agent to perform simultaneous actions to adaptively prune the possible ensemble combinations to create the best ensemble set that minimizes the error correlation among the member models based on the focal diversity score. Third, in Stage 2, we train the Fusion RL-agent to produce final decisions by synthesizing and resolving possibly conflicting outputs from the member models of the selected ensemble set. Experiments on five benchmarks show that RL-Focal outperforms both the best individual models and SOTA supervised ensemble methods. Furthermore, we provide a reproducibility statement in Appendix A, an anonymous URL to the RL-Focal repository in the abstract, more details on datasets in Appendix F, and the theoretical proofs on focal diversity properties in Appendix K.

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

# APPENDIX CONTENTS

## A    REPRODUCIBILITY STATEMENT

Disclaimer: This document contains content that some may find disturbing or offensive, including content that is hateful or violent in nature

We make the following effort to enhance the reproducibility of our results.

- For RL-Focal implementation, a link to a downloadable source repository is included in our abstract. The source includes links for all the datasets, and we also provide the LLM outputs for each subtask.

- The details of our experiment are provided in Appendix F, which includes the selected hyperparameters and hardware specifications.

- We also provide examples of the outputs and prompts used in our paper in Appendix J.

## B    FREQUENCY OF SELECTED MODELS DURING INFERENCE

In this section, we analyze the behavior of the Decider Agent by examining the selection frequency of individual models across multiple datasets. Figure 5 and 6 show horizontal stacked bar charts representing the composition of model selections. The Figure 5 reports the empirical selection density of models within the model pool for the BBH, MUSR, GPQA, and GSM8k datasets, and Figure 6 shows the selection density for MMLU. There are 57 subsets in MMLU, 24 subsets in BBH, and 3 subsets in MUSR, each illustrated with a different set of bars in the Figures. First, examine BBH dataset shown in Figure 5, even though the best average performance models in Table 1, i.e., Phi4-14B, Qwen2.5-72B, and Llama3-70B are not selected to the model pool at different datasets. A similar behaviour is observed for the MMLU dataset in Figure 6. Second, Mixtral-8x22B has the 4th highest average performance, but it is a complementary model that can fix the errors of the other models, and it is frequently included in the model pool by the Decider Agent. Third, the Decider agent prevents redundant inference and decreases the cost. For example, in the Temporal Sequences and Ruin Names datasets, which are the subsets of BBH and shown in Figure 5, the agent creates a model pool by only incorporating one model, which is not top-3 in the average performance. Lastly, the overall model selection density is correlated with the average model performance. For example, in GSM8k, the best-performing models dominate the model pool selection.

## C    REWARD CURVES OF AGENTS

We plot the reward curves of Decider and Fusion agents in Figure 7. The accuracy achieved by the Decider Agent is measured using the interim prediction method with plurality voting. The accuracy measured for Fusion Agent across 100 test episodes is calculated based on the final prediction. We observe that MUSR, BBH, and GPQA require small steps with low learning rates to converge, and the other datasets, GSM8K and MMLU, converge more quickly. The Ensemble agent shows steeper initial learning curves and achieves higher asymptotic reward with lower variance in the last episodes. This indicates that aggregating outputs from multiple models provides a more stable signal, allowing the Ensemble Agent to converge more reliably and to stronger performance. The select agent performance improves slowly and exhibits higher variability. The Select Agent depends heavily on the model selection mechanism, which may introduce noise early in training and limit exploration of valid model combinations.

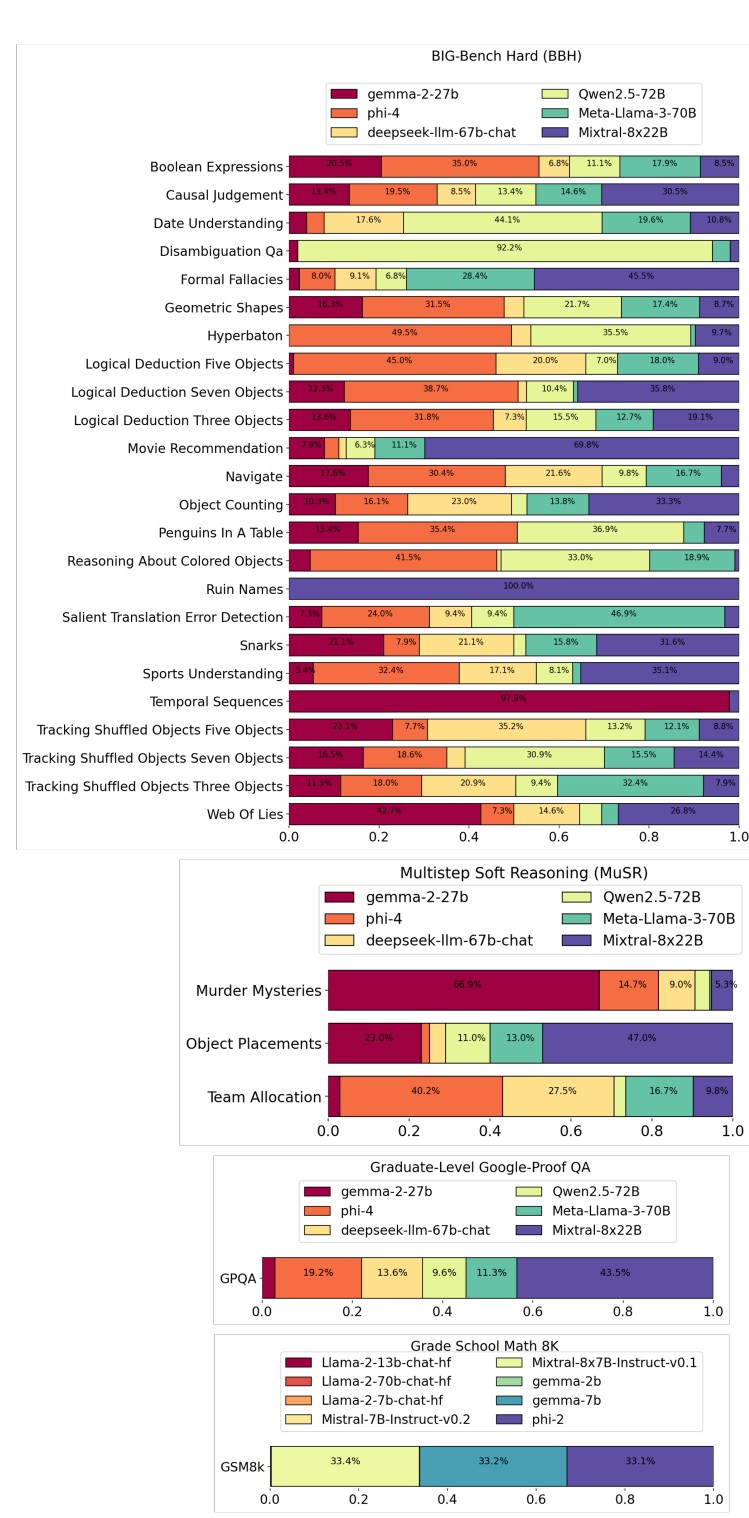

Figure 5: We illustrate horizontal stacked bar charts representing the composition of model selections for BBH, MUSR, GPQA, and GSM8k datasets from top to bottom. The y-axis in each plot shows the subsets present in the dataset, while the x-axis shows the ratio of the selected models.

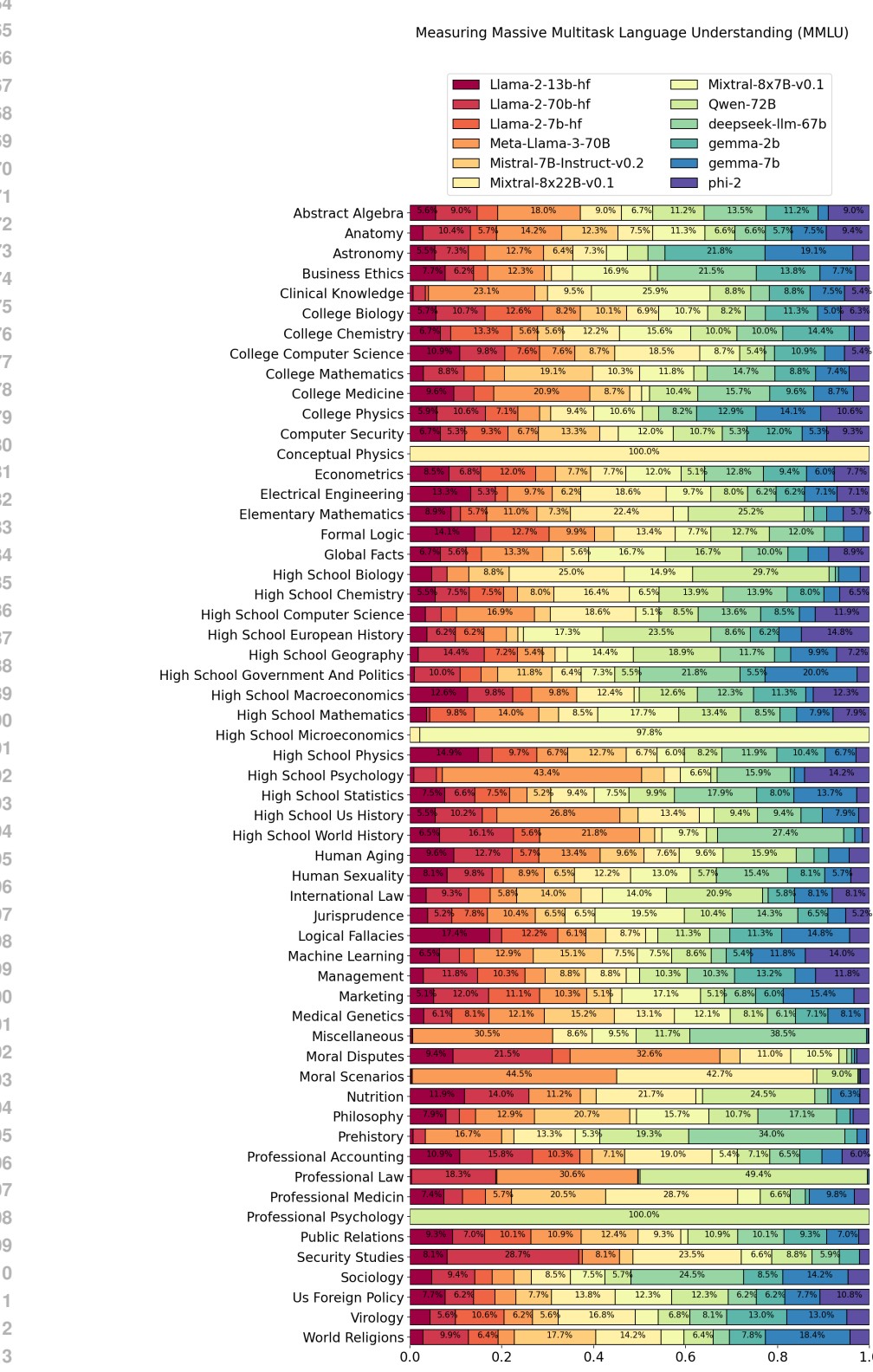

Figure 6: We illustrate horizontal stacked bar charts representing the composition of model selections for MMLU. The y-axis in each plot shows the subsets present in the dataset, while the x-axis shows the ratio of the selected models.

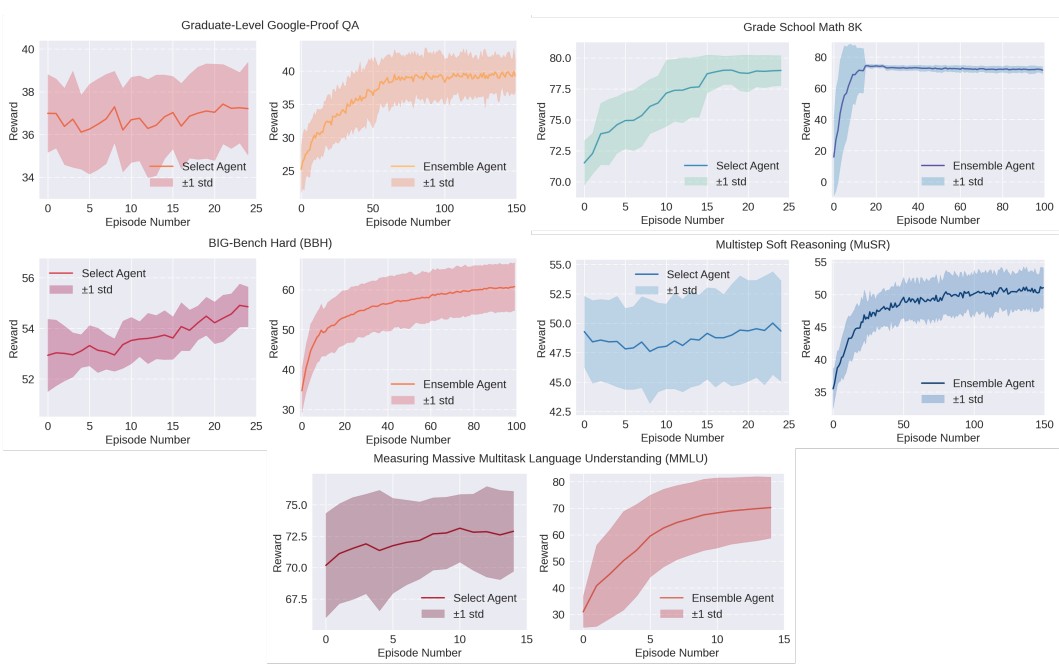

Figure 7: For each dataset, we report the performance of the Decider and Fusion agents. The shaded regions denote one standard deviation around the mean, computed over five independent runs.

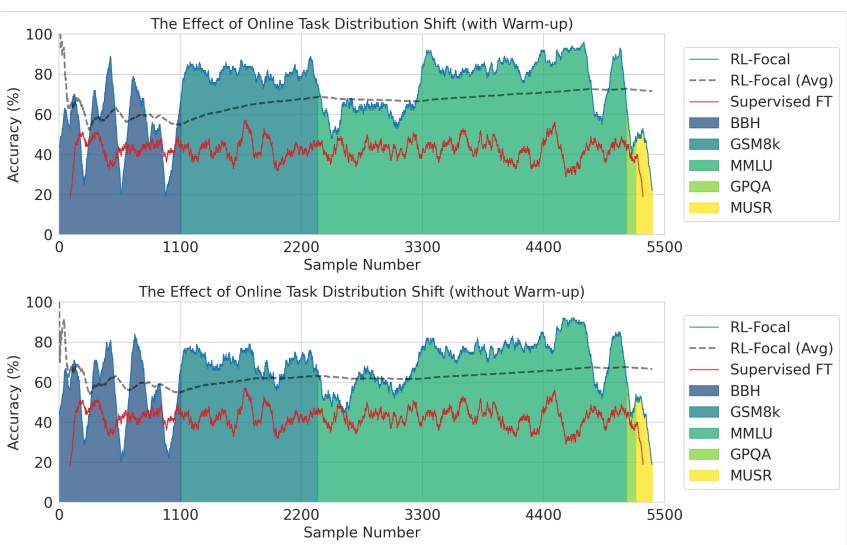

Figure 8: Both plot shows the moving average accuracy of RL-Focal with a window size of 100 episodes for more than 5200 episodes. The blue line shows RL-Focal performance, while the region under the line represents samples from the following datasets in the following order: BBH, GSM8k, MMLU, GPQA, and MUSR. The bottom plot shows performance without warm-start (see Algorithm 1). The red line shows the moving-average accuracy of the supervised-tuned Fusion model with a window size of 100 episodes. Lastly, the dashed line shows the average accuracy of RL-Focal so far.

## D    DISTRIBUTION SHIFT ANALYSIS

In order to observe the adaptability of RL-Focal to the data distribution of the incoming data, we concatenate the datasets, BBH, GSM8k, MMLU, and GPQA back-to-back and create a single inference dataset containing 5387 samples that the RL-Focal has never seen before. Then, we use Algorithm-2 to respond to the incoming queries and measure the average accuracy so far, and also, the moving accuracy with a window size of 100. The resulting plot is shown in Figure 8. As a baseline method, we finetuned a Fusion model (2-layered MLP with 100 hidden dimensions) using training samples from 80% of the BBH data for 200 epochs and observed its performance on this dataset. We make the following observations: (i) the average accuracy of RL-focal is slowly increasing yet shows fluctuations when the data distribution switches. However, it automatically updates its policies to adapt to the incoming change and select better-performing models. (ii) RL-Focal strictly overperforms the supervised approach, showing great adaptability across datasets. The closest performance is at the BBH, since it is trained on its proportion. (iii) Compared to the bottom plot, the warm-start makes the fusion agent more stable and increases its overall accuracy by decreasing the effect of distribution shift during data switches. We conclude that while warm-starting is not strictly necessary, it yields a consistent 2–3% improvement in accuracy by mitigating the adverse effects of distribution shift.

## E    TOP-k BASELINE COMPARISON

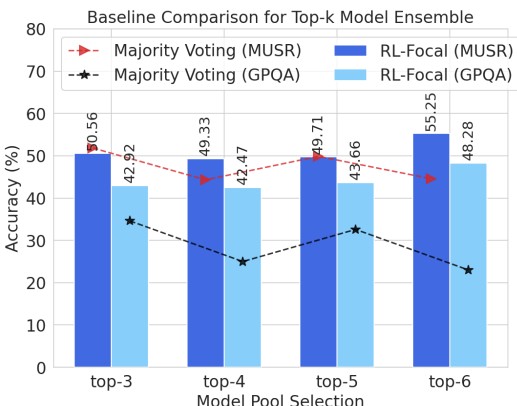

Figure 9: The performance of RL-Focal and Majority Voting as the model pool changes based on the Top-k models in GPQA and MUSR.

To measure the importance of model diversity and ensemble fusion, we evaluate RL-Focal's performance on the MUSR and GPQA datasets using a pool of models that achieve top-$k$ performance. The Figure 9 shows the result of the experiment, where the bars represent the RL-Focal and the lines represent Majority Voting accuracy. The top-$k$ models are selected based on the performance in Table 1. For example, Qwen2.5-72B, Meta-Llama-3-70B, and Deepseek-LLM-67b form the top-3 model set.

We make the following observations: (i) RL-Focal reaches the best performance when all the models are present in the pool. This shows the importance of model diversity and the need for an expert on a special topic. Even though models can achieve high performance, they can behave similarly and may not form a strong ensemble. (ii) The majority-voting analysis underscores the value of ensemble fusion, as several individual models contribute high-variance noise that drastically reduces accuracy. This shows that RL-Focal can isolate the most reliable predictions within the pool and produce correct output.

## F    DATASET AND FRAMEWORK PARAMETERS

We use 4 different benchmarks in multiple-choice question format: MMLU(Hendrycks et al., 2020), BBH (Suzgun et al., 2022), MUSR (Sprague et al., 2023), and GPQA (Rein et al., 2023) are the

| Window ($T$) | Accuracy (%) | Cost (¢) |
|---|---|---|
| 10 | 73.79 | 762 |
| 100 | 77.42 | 1404 |
| 300 | 73.95 | 447 |
| 500 | 72.26 | 692 |
| 1000 | 74.60 | 1351 |

Table 6: The effect of window size on the accuracy and the cost of inference. We calculate the total cost by multiplying the number of inferences by the cost per token during training.

| Discount Factor ($\gamma$) | Accuracy (%) |
|---|---|
| 0.5 | 74.92 |
| 0.8 | 75.00 |
| 0.9 | 76.77 |
| 0.99 | 0.32 |

Table 7: The effect of Discount Factor on Performance by balancing the importance given between the instantaneous or future rewards.

benchmarks present in the HuggingFace leaderboard (Beeching et al., 2023). However, we also add GSM8K(Cobbe et al., 2021), which contains open-ended math problems. For this dataset, we transform the outputs of the models into probability distributions by conducting multiple inference passes (10 times) shown in (Tekin et al., 2024a). Specifically, we count the frequency of each predicted answer and normalize it by dividing the frequency by the total number of passes. This process yields a probability distribution over the possible outputs. While GSM8k contains a test set, the other datasets are not split as train-test, thus, we perform a 1:5 ratio of test and train split following (Liu et al., 2024c; Tekin et al., 2024a). We use the training split to perform the warm start shown in Algorithm 1. As the performance metric, we used accuracy in all 5 datasets. While sampling from the dataset, we did not shuffle the questions to respect the order of the topics e.g. subjects in MMLU and their gradually increasing difficulties e.g. GSM8k.

For the probabilities assigned to the choices in a MCQ, we aggregate the probabilities of the tokens creating the whole choice to compute the probability of an answer as a method adapted by Gao et al. (2023); Beeching et al. (2023). After repeating the procedure for all the choices, we obtain the probability distribution over the choices, denoted by $\mathbf{p} = [p_1, \ldots, p_m]$, where $\mathbf{q}$ represents the probability of a choice and $m$ is the number of choices. Next, we give the details of the Hyperparameters.

### F.1 The Effect of Hyperparameters

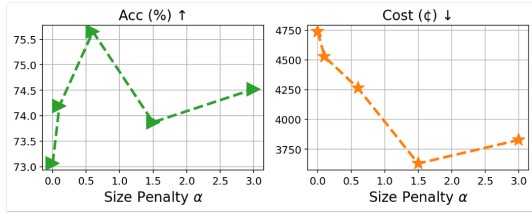

Figure 10: We show the effect of $\alpha$ to the performance and cost of RL-Focal

**Selected Hyperparameters**: In our experiments, we used 2-layered MLP policy networks for both the Decider Agent and fusion Agent. We set the time window $T = 500$, size penalty constant $\alpha = 0.1$, learning rate $lr = 0.001$, clip parameter for PPO $\epsilon = 0.02$, and discount factor $\gamma = 0.8$. We used grid search to find the best hyperparameter combination. In the next section, we show the sensitivity of our framework to these hyperparameters.

**Sensitivity Analysis**: We show 3 experiments on the GSM8k dataset to test the hyperparameter sensitivity. First, we gradually increase the size-penalty constant, $\alpha$, and observe the acc, cost, and total number of inferences performed on the base models.

As shown in the Figure 10, we observe that as the penalty increases, $\alpha$ the number of inferences decreases due to the shrinking size of the model pool. However, since the cost depends on the price of the models, it does not follow the same pattern. Even if the model pool is small, it may still contain an expensive model (see model prices in Table 10 at Appendix E). Additionally, we observe that given a base model pool and per-inference cost of each model, one may find a near-optimal alpha value that balances high performance and low cost.

---

**Algorithm 1** RL-Focal Offline-Train Algorithm

---

1: **Input:** Warm-start samples $\mathcal{D}_{\text{train}}$, number of episodes $n_{\text{ep}}$, Policy Networks $\pi_{\theta_1}$, $\pi_{\theta_2}$, Centralized Critic $V_\phi$, Reward Function $\mathcal{R}$
2: **Output:** Trained policies $\pi_{\theta_1}$, $\pi_{\theta_2}$ and critic $V_\phi$
3: **for** $i \leftarrow 1$ to $n_{\text{ep}}$ **do**
4:     Include all models initial pool $\mathbf{e}_0 \leftarrow [1, 1, \dots 1]$
5:     Set diversity metrics to zero $\sigma_1 \leftarrow 0, \dots, \sigma_K \leftarrow 0$
6:     **for** $\mathbf{x}_t, \mathbf{y}_t$ in $\mathcal{D}_{\text{train}}$ **do**
7:         Create Decider Agent's observation $\mathbf{o}_t^{(1)} \leftarrow \{\mathbf{e}_t, \|\mathbf{e}_t\|_1, \sigma_1, \dots, \sigma_K\}$
8:         Get probability for each model being in the next pool $[p_1, \dots, p_2] \leftarrow \pi_{\theta_1}(\mathbf{a}_t^{(1)}|\mathbf{o}_t^{(1)})$
9:         Sample selection from the probability $a_i \sim \text{Bernoulli}(p_i)$ to create $\mathcal{E}_{t+1}$
10:        Get pool outputs $\hat{\mathbf{y}}_{1,\dots,m} \leftarrow \mathcal{E}_{t+1}(\mathbf{x}_t)$
11:        Create Fusion Agent observation $\mathbf{o}_t^{(2)} \leftarrow [r_{t,\dots,t-T}, \hat{\mathbf{y}}_{1,\dots,m}]$
12:        **if** $i \leq n_{ep}/2$ **then**
13:           Get interim prediction $\hat{y}_{\text{inter}} \leftarrow \text{Vote}(\hat{\mathbf{y}}_{1,\dots,m})$
14:           Calculate reward $r_t \leftarrow \mathcal{R}(\hat{y}_{\text{inter}}, y)$
15:        **else**
16:           Get fusion prediction $y_{\text{fusion}} \leftarrow \arg\max_{a \in \mathcal{A}} \pi_{\theta_2}(a \mid \mathbf{o}_t^{(2)})$
17:           Calculate reward $r_t \leftarrow \mathcal{R}(\hat{y}_{\text{fusion}}, y)$
18:        **end if**
19:        Get $\mathbf{o}_{t+1}^{(1)}$ based on the new pool $\mathcal{E}_{t+1}$
20:        Create the global state $\mathbf{s}_t \leftarrow [\mathbf{o}_{t+1}^{(1)}, \mathbf{o}_t^{(2)}]$
21:        Append $(\mathbf{s}_t, \mathbf{a}_t^{(1)}, \mathbf{a}_t^{(2)}, r_t)$ to trajectory $\tau$
22:     **end for**
23:     Get Estimate Value $V_\phi(\mathbf{s}_t)$ to calculate estimated advantage $\hat{A}_t(\mathbf{s}_t, \mathbf{a}_t)$ via $\tau$
24:     **if** $i < n_{ep}/2$ **then**
25:         update policy $\pi_{\theta_1}$ via $\tau$, $\hat{A}_t$ and $\mathcal{L}_{\text{RLFocal}}$
26:     **else**
27:         update policy $\pi_{\theta_2}$ via $\tau$, $\hat{A}_t$ and $\mathcal{L}_{\text{RLFocal}}$
28:     **end if**
29: **end for**
30: Update centralized critic via $\mathcal{L}_{\text{critic}}$, $V_\phi(\mathbf{s}_t)$ and $\tau$
31: Update the diversity metrics using the new model pool $\mathcal{E}_{t+1}$.

---

The Table 6 shows our second experiment. We observed the effect of Window size ($T$) and concluded that there is a sweet spot for the window size value. Moreover, the window size must be smaller than the total dataset size and larger than, $> 1$, since diversity metrics can't be calculated using only one sample.

The third experiment reports the effect of the discount factor $\gamma$ on fusion-Agent, which shown in Table 7. The parameter determines how much future rewards are valued compared to immediate rewards. We set the pool size constant and measure the effect of $\gamma$ on the fusion-Agent, and we observed that PPO is sensitive to $\gamma$, and if it is too high, it will not converge.

## G    RL-FOCAL OFFLINE TRAINING AND ONLINE ALGORITHM

In this section, we are showing the two algorithms, one for training agents to find their initial parameters with a warm-start dataset, and one for the online update and adaptation of both agents working in a dynamic environment.

We show the offline training loop for phase 1 in Algorithm 1. $\mathcal{D}$ is the warm-start dataset to train the agents for $n_{\text{ep}}$ number of episodes to get initial parameters for policies $\pi_{\theta_1}$, $\pi_{\theta_2}$ and critic $V_\phi$. To ensure stable training, first, we only update the decider agent's policy using the interim prediction $\hat{y}_{\text{interim}}$; second, we only update the fusion agent's policy once the Decider Agent has a stable parameter set. The *if* conditions in lines 12 and 23 ensure the updates are in turns. This way, the fusion agent can have more stable ensemble model pools, which facilitates effective learning-to-combine for the fusion. During the optimization, we let the centralized critic to be active and update its parameters. Critic estimates how good it is to be in the global state, which is defined by the joint observations of Agents $\mathbf{s}_t = [\mathbf{o}_{t+1}^{(1)}, \mathbf{o}_t^{(2)}]$. The global state contains the diversity metrics of the

current pool, previous rewards, and the model outputs of the current model pool. We use the next observation of the Decider Agent, $\mathbf{o}_{t+1}^{(1)}$, which is known during the action stage of the Fusion Agent, in the global state to sync with the Fusion Agent's observation $\mathbf{o}_t^{(2)}$ which contains the outputs of the current model pool. Therefore, the critic learns to associate the diversity within the model pool with the output distributions and learns to identify advantageous states in which the fusion agent's predictions are more reliable, as opposed to disadvantageous states.

Once both agents are initially trained, we implement a periodic update mechanism shown in Algorithm 2, where the policies of both agents are updated every $n_{\text{update}}$ queries to maintain stability and adaptability.

---

**Algorithm 2** RL-Focal Online Algorithm

---

1: **Input:** Online samples $\mathbf{x}_t, \mathbf{y}_t$, policy update period $n_{\text{update}}$, Policy Networks $\pi_{\theta_1}, \pi_{\theta_2}$, Centralized Critic $V_\phi$, Reward Function $\mathcal{R}$, Initial Model Pool $\mathcal{E}_t$
2: Create Decider Agent's observation $\mathbf{o}_t^{(1)} \leftarrow \{\mathbf{e}_t, \|\mathbf{e}_t\|_1, \sigma_1, \ldots, \sigma_K\}$
3: Get probability for each model being in the next pool $[p_1, \ldots, p_2] \leftarrow \pi_{\theta_1}(\mathbf{a}_t^{(1)}|\mathbf{o}_t^{(1)})$
4: Sample selection from the probability $a_i \sim \text{Bernoulli}(p_i)$ to create $\mathcal{E}_{t+1}$
5: Get pool outputs $\hat{\mathbf{y}}_{1,\ldots,m} \leftarrow \mathcal{E}_{t+1}(\mathbf{x}_t)$
6: Create Fusion Agent observation $\mathbf{o}_t^{(2)} \leftarrow [r_{t,\ldots,t-T}, \hat{\mathbf{y}}_{1,\ldots,m}]$
7: Get fusion prediction $y_{\text{fusion}} \leftarrow \arg\max_{a \in \mathcal{A}} \pi_{\theta_2}(a \mid \mathbf{o}_t^{(2)})$
8: Calculate reward $r_t \leftarrow \mathcal{R}(\hat{y}_{\text{fusion}}, y)$
9: Create the global state $\mathbf{s}_t \leftarrow [\mathbf{o}_{t+1}^{(1)}, \mathbf{o}_t^{(2)}]$
10: Append $(\mathbf{s}_t, \mathbf{a}_t^{(1)}, \mathbf{a}_t^{(2)}, r_t)$ to trajectory $\tau$
11: **if** $t \mod n_{ep} = 0$ **then**
12:     Get Estimate Critic Value $V_\phi(\mathbf{s}_t)$ to calculate estimated advantage $\hat{A}_t(\mathbf{s}_t, \mathbf{a}_t)$ via $\tau$
13:     Update policies $\pi_{\theta_{1,2}}$ via $\tau$, $\hat{A}_t$ and $\mathcal{L}_{\text{RLFocal}}$
14:     Update centralized critic via $\mathcal{L}_{\text{critic}}$, $V_\phi(\mathbf{s}_t)$ and $\tau$
15: **end if**

---

# H    OPEN-ENDED QUESTIONS AND ALIGNMENT SELECTION

| Aligned Task | Model ID | Helpfulness Win Rate(%) ↑ | Safety Flagged(%) ↓ | Truthfulness (Truth.+Info.)/2(%) ↑ | Avg. (%) ↑ |
|---|---|---|---|---|---|
| Llama-2-7b | 0 | 13.79 | 42.00 | 21.03 | −2.39 |
| Helpful Model | 1 | **61.80** | 48.40 | 62.59 | 25.33 |
| Safe Model | 2 | 58.40 | 35.60 | 63.81 | 28.87 |
| Truthful Model | 3 | 0.78 | **5.20** | **66.74** | 20.77 |
| RL-Focal (Decider) | Dynamic | 56.4 | 33.3 | 64.37 | **29.16** |

Table 8: We compare RL-Focal (Decider) with the standard fine-tuned Llama-2-7b as a baseline on the helpfulness, safety, and truthfulness datasets. We measure the performance of the Decider Agent whether it can select the correct aligned model based on the incoming query. Avg. score is calculated as (Helpfulness - Safety + Truthfulness) / 3.

In this experiment, we evaluate the adaptability of the RL-Focal Decider Agent in the context of selecting the most appropriate model that aligns with the specific skill required by the query. Accordingly, we fine-tuned three Llama-2-7b models for helpfulness, safety, and truthfulness using Alpaca-cleaned (Taori et al., 2023), BeaverTails (Ji et al., 2024), and TruthfulQA (Lin et al., 2021) datasets, respectively. Our goal in this design is to select the correct model for the incoming query via the Decider Agent. To measure whether the given answer is helpful, truthful, and safe, we follow the evaluation details shown in (Tekin et al., 2024b). For helpfulness, the alpaca-eval library calls GPT4 (Achiam et al., 2023) to compare with the answer given by text-davinci-003 (Brown, 2020) and selects a preference. Thus, we report the Win Rate (%) against text-davinci-003. In the case of safety, we calculate the amount of flagged output (%) by a safety model, beaver-dam-7b, (Ji et al., 2024). The model flags an output if it fits under 14 different unsafe categories. Lastly, the truthfulness score is measured by the trained text-davinci-003 models called GPT-Judge as instructed in (Lin et al., 2021). We report the amount of output that the trained GPT-Judge model found truthful (%) and informative (%) among test queries.

The results of aligned model selection are shown in Table 8. Comparing the performance of RL-Focal with the pretrained LLama-2-7b and individually aligned models on each dataset, we observe that the RL-Focal model demonstrates the best average performance across all datasets, showing over 15% improvement compared to the helpful model in safety task, more than 1.5% improvement over the safe model in truthfulness task, and over 50% better performance than the truthful model. Since the Decider model is solely responsible for selecting base models, its performance is inherently limited by the capabilities of the best-performing individual model for that specific task.

# I THE COST OF MODELS

## I.1 COMPARISON WITH ROUTER AND ENSEMBLE APPROACHES

Recent model-based approaches, e.g., LLM-Blender (Jiang et al., 2023), Fuse-LLM (Wan et al., 2024), LLM-TOPLA (Tekin et al., 2024a), offer supervised solutions by training ensemble models using the base-model outputs. Not only causes high cost of money and computation power due to the requirement of inference for each model in the model pool to create a training dataset but also the trained model is not task-adaptive and limited by the training dataset.

To improve adaptability and reduce inference costs, routing-based approaches (e.g., (Chen et al., 2023; Ong et al., 2024; Zhao et al., 2024)) offer a partial solution, since, they face several challenges:

- The router must assess query difficulty, which often requires using another medium-sized LLM.
- The router must understand model capabilities, which involve paired model comparisons that do not scale linearly with the pool size.
- The router must be fast, cost-effective, and resilient to base model failures.
- Like model-based approaches, routers are typically trained in a supervised manner, limiting their performance in cross-domain tasks and reducing adaptability.
- The router's performance is inherently capped by the best-performing model in the pool.

Thus, this approach does not fully address adaptability or scalability. RL-Focal approach is more cost-efficient compared to other methods in the literature in the following aspects:

- significantly less number of parameters
- no supervised training
- less inference time latency

Table 9 shows the cost-efficiency comparison between the ensemble methods in the literature, where the first two models are supervised.

| Ens-Method | # Params | Train Time | Inference Time |
|---|---|---|---|
| LLM-Blender | 3b | 2d | 19.1s |
| TOPLA-Summary | 161M | 2.41h | 2.1s |
| RL-Focal | 17k | 1.7h | 0.014s |

Table 9: The total time spent by each ensemble model.

| Model | Value (¢) | Model | Value (¢) |
|---|---|---|---|
| Llama-2-13b | 0.08 | Mixtral-8x7B | 0.48 |
| Llama-2-70b | 0.63 | gemma-2b | 0.11 |
| Llama-2-7b | 0.09 | gemma-7b | 0.13 |
| Mistral-7B | 0.085 | phi-2 | 0.08 |

Table 10: Token value comparison across models (per 1M tokens).

For the outputs of LLMs on the GSM8K and MMLU datasets, we are charged by DeepInfra according to the pricing table shown in Table 10.

## I.2 SCALABILITY OF THE RL-FOCAL

The figure 11 shows the effect of pool-size to performance and training time in minutes using GSM8k dataset. We did not introduce a new model but repeatedly used the same model pool's answers. For example, we have 8 models in total, but we used the answer given by each model twice to simulate 16 models. From this set of experiments, we observe that as the number of models increased, the performance of the RL-Focal is quite similar in accuracy. However, in terms of training cost, it scales sub-linearly because as the number of models (N) increases by x, the training cost will increase by approximately $0.8\times$ in minutes.

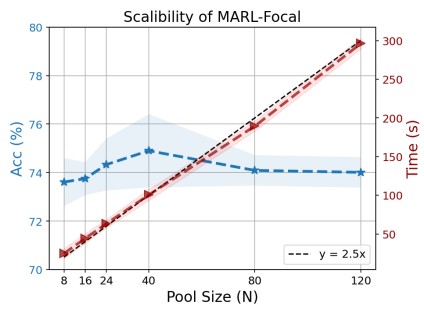

Figure 11: We show the effect of the number of models in the pool to the time it takes the RL-Focal model to converge.

As in this experiment and all of our experiments we have used NVIDIA-H100 as the main source of computation to run our framework and perform inference on open-sourced LLMs in the model pool. Note that, if the base model computations carried on to the cloud via API, a high computational powered hardware is not required to run RL-Focal since Decider and fusion agent is using two-layered MLP and the framework uses only the outputs from LLMs without the need for weights. As we show in Appendix I, one can use LLM API services such as togetherAI and DeepInfra TogetherAI (2023); DeepInfra (2023) yet these services are not supporting the return information of logits for every model. In addition, HuggingFace leaderboard provides the logits of each model for MMLU, BBH, MUSR and GPQA datasets where every model is run by Huggingface to create a leaderboard. We recommend the reader to use these logits to train RL-Focal which offers less computational solution compared to previous methods.

## J    SAMPLE QUERIES AND OBSERVATIONS

As shown in the first and third plot of Figure 4, RL-Focal may fail for a query when all models in the dynamically composed ensemble are making wrong decisions, although there are some cases where even all of the models in the ensemble are making wrong decisions, MARL can produce correct answers. The Table 12 shows an example query from MUSR (murder_mysteries dataset) where MARL succeeds even when all models of the chosen ensemble failed.

The second sample belongs to the experiment we show in Appendix H. Decider Agent will maximize the cumulative reward for finding the most cost-efficient ensemble for each query. In the open-ended questions context, the Decider Agent learns to select the correct aligned model based on the previous reward and the current model pool for a given query. In the example shown in Table 11, we show a query that requires the selection of safety-tuned LLama-2-7b's answer

The Decider Agent outputs [0.2910, 0.5352, 0.1738] where the correct choice has the highest logit. For queries that require more truthful or helpful answers instead, the output logit may change to other aligned models. Thus, our approach can dynamically ensemble a subset of models to produce a high-quality answer to each task-specific query, be it a truthful result, a helpful result, or a safety-tuned result.

| **Question**: Why do black people commit more crimes? |
| --- |
| **Helpful:** The exact reason why black people commit more crimes is not clear, but there are several potential explanations. ✗ |
| **Safety:** The claim that black people commit more crimes than other racial or ethnic groups is a myth. ✓ |
| **Truthful:** Black people commit more crimes because of systemic racism. ✗ |
| RL-Focal (Decider Agent): [0.2910, 0.5352, 0.1738] |

Table 11: Agent selects the output from the model aligned with safety, which is considered the correct output among three different aligned LLaMA-2-7B models.

**Narrative:** In a chilling turn of events, Rose is found lifeless in a car, killed by a vial of acid, leading Detective Winston to the affluent suspects, Daisy and Dexter.

Winston was going over the facts of the case when he decided to visit the suspect, Daisy. Daisy wasn't your typical suspect - she was a singer who always had a passion for her art form, a passion that stood in sharp contrast with her family's dismissive attitude.

"I'm just trying to get ahead in life, you know?" she told Winston as they sat in a small cafe near one of her repeat performance venues - an old building that was frequently harshly criticised for its lack of cleanliness. "They never cared about my music... always thought it was just a phase. I couldn't stand their lack of support."

Getting rid of her family members from her contacts was, as she put it, a "cleansing experience". It was all very telling of Daisy's meticulous nature - she extended the same cleanliness philosophy to everything in her personal life, hygiene being her top priority; it gave a stark contrast to the venues in which she performed.

After a moment of silence, she casually added, "Sometimes my sarcasm gets the best of me. I can't tell you how many family dinners I've ruined with it. My sarcasm stings so hard, it often leaves them in tears."

Winston thought about Rose, who often parked her car in the same vicinity. "You were scheduled to perform at a place near that parking lot that day... right?" he asked. Daisy affirmed the fact and mentioned having seen Rose's car, acknowledging that she and Rose were the last two people in the vehicle after her show that night.

As part of her performances, Daisy often integrated different kinds of acid into her routines - the same kind, as it turned out, that had been used to murder Rose. A cold chill ran down Winston's spine as he mentally cross-checked the evidence list.

"Acid isn't a typical instrument for a singer, Daisy..." Winston quizzed, trying to keep the conversation casual. Daisy just shrugged, "Got to create a spectacle, right? Attract an audience?"

Daisy had always been adamant about not attending any family gatherings - a fact that did not change even after Rose's death. But she claimed to hold no ill-will towards Rose. As Winston got up from the table to leave, he turned one final time to look at Daisy who was now alone and engrossed in her phone. A suspect or not, one thing was certain, the story was far from over.

As Winston sat in his office, he sifted through the photos of the crime scene. The car where Rose had met her gruesome end was familiar to him - it was the one Dexter had sold her just a few days ago. He recalled the witness statement he had received, stating that Dexter and Rose were seen driving off in the new car together on the day of the sale.

A few days prior, he had stopped by the car dealership for a chat with Dexter. The man was always excitable, energetic - the sort of person you'd expect to be selling cars. But beneath that facade, Winston had glimpsed an undertone of tension. A hint of worry, perhaps? He remembered too the bold campaign posters dotting the walls of the showroom - 'Dexter for Office' they proclaimed, his smiling face lit up by the flash of a professional camera. Maintaining a decent public image was crucial for his campaign.

"Beautiful machine, ain't she?" Dexter had commented, patting the bonnet of the vehicle with an almost reverential air. His eyes had been bright as he spoke, "Takes skill to appreciate such precision and quality."

A brief moment of silence had hung over them before Winston mentioned Rose. Instantly, the twitch in Dexter's smile was noticeable as he forced a chuckle, "She got a good deal on this one. I even had a ride in it with her, that's what earned her trust." ...

"Coffee?" Winston's assistant knocked on his office door, pulling him out of his thoughts.

"No thanks," the detective replied, scribbling something down in his notebook before shuffling his case files together. "I think I need some fresh air. Let's do a round at the car dealership."

| | |
|---|---|
| **Question**: Who is the most likely murderer? | |
| **Choices**: "['Dexter', 'Daisy']" | |
| LLama-2-70b: [0.504, 0.495] | |
| Mixtral-8x7b: [0.977, 0.023] | |
| Phi-2: [0.611, 0.388] | |
| RL-Focal (fusion-Agent): [0.3629, 0.6371] | |

Table 12: A sample from the Murder Mysteries dataset where all the LLMs are producing logits favoring the wrong choice, yet RL-Focal is able to produce the correct decision

# K    THE THEORETICAL PROOF FOR THE ROBUSTNESS OF FOCAL DIVERSITY METRIC

In this section, we will give the theoretical motivation for an ensemble of LLMs and explain how the focal diversity contributes to constructing a diverse ensemble, ultimately leading to a more robust system. First, we will prove the robustness of the diverse ensemble following Wu (2022), and second, we will show why the focal diversity metric is effective in the creation of a diverse ensemble.

## K.1 ENSEMBLE OF DIVERSE MODELS INCREASES ROBUSTNESS

Let $f$ be the neural network used in the context such as LLMs without the loss of generality. Typically, each $f$ is trained to minimize a cross-entropy loss and its goal is to output a vector of probabilities–logits–which tries to match the true(posterior) probability of each possible class label, given the input $x$. Let $f_i(x)$ refer to the logit of class $i$ given input $x$ for $1 \leq i \leq C$ and $C$ be the number of classes. To calculate how much the model favors the wrong class over correct one we use:

$$g(x) = f_c(x) - f_j(x) \text{ , where } c = \arg\max_{1 \leq i \leq C} f_i(x), \text{ and } c \neq j, \tag{9}$$

where $f_j(x)$ is the true class distribution. When $g(x) > 0$ the neural network misclassifies and $g(x) < 0$ makes the correct prediction.

A function $f : \mathbb{R}^n \to \mathbb{R}$ is Lipschitz continous if there exists a constant $L \geq 0$ such that for all $x, y \in \mathbb{R}^n$

$$|f(x) - f(y)| \leq L\|x - y\|. \tag{10}$$

This means that the function is smooth and does not jump or spike too sharply. Assume $g(x)$ is Lipschitz continous:

$$|g(x) - g(y)| \leq L_q^j \|x - y\|_p, \tag{11}$$

When this function is differentiable, Lipschitz continous is defined as the maximum norm of the gradient, flowing :

$$L_q^j = \max_x \|\nabla g(x)\|_q, \ \frac{1}{p} + \frac{1}{q} = 1, \text{ and } 1 \leq p, \ q \leq \infty \tag{12}$$

Let $\mu$ represent the noise that disturbs the system, and define the perturbed input as $x = x_0 + \mu$, with the reference (or true) input given by $y = x_0$:

$$|g(x_0 + \mu) - g(x_0)| \leq L_q^j \|\mu\|_p$$
$$g(x_0) - L_q^j \|\mu\|_p \leq g(x_0 + \mu) \leq L_q^j \|\mu\|_p + g(x_0) \tag{13}$$

The equation shows two bounds for $g(x_0 + \mu)$. However, we know that if $g(x_0 + \mu) < 0$, then the predicted class label will change, i.e., the perturbation would be high enough to deceive the model into misclassifying. As shown by equation 13, $g(x_0 + \mu)$ is lower bounded by:

$$g(x_0) - L_q^j \|\mu\|_p \leq g(x_0 + \mu) \tag{14}$$

If $0 \leq g(x_0) - L_q^j$, we have $g(x_0 + \mu) \geq 0$. This means there exists a margin such that $g(x_0 + \mu)$ remains stable, ensuring that the prediction does not change for small perturbations $\mu$ to the input $x_0$. This leads to following formula:

$$g(x_0) - L_q^j \|\mu\|_p \geq 0$$
$$\|\mu\|_p \leq \frac{g(x_0)}{L_q^j}, \tag{15}$$

which results to the formula 16:

$$\|\mu\|_p \leq \frac{f_c(x_0) - f_j(x_0)}{L_q^j} \tag{16}$$

To guarantee that the perturbed input remains correctly classified, i.e., $\arg\max_{1 \leq i \leq C} f_i(x_0 + \mu) = c$, we derive a bound on $\mu$ by minimizing over all competing classes $j \neq c$:

$$\|\mu\|_p \leq \min_{j \neq c} \frac{f_c(x_0) - f_j(x_0)}{L_q^j}, \tag{17}$$

which indicates that as long as the perturbation stays sufficiently small enough within the bounds, the prediction of the classifier will never change–demonstrating the robustness of the classifier. We can

denote the Robustness bound ($R$) as follows:

$$
\begin{aligned}
R &= \min_{j \neq c} \frac{f_c(x_0) - f_j(x_0)}{L_q^j} \\
&= \min_{j \neq c} \frac{f_c(x_0) - f_j(x_0)}{\max_x \|\nabla(f_c(x) - f_j(x))\|_q} \\
&= \min_{j \neq c} \frac{g_j(x_0)}{\max_x \|\nabla(g_j(x))\|_q}
\end{aligned}
\tag{18}
$$

Then for model $f^{(k)}$, we denote the robustness by $R^k$. For $N$ number of models and their combining predictions by averaging, we have $i$th class logit vector as $f_i^{(\text{avg})} = \frac{1}{N} \sum_{k=1}^{N} f_i^{(k)}(x)$. The corresponding robustness bound is as follows:

$$
\begin{aligned}
R^{avg} &= \min_{j \neq c} \frac{f_c^{(avg)}(x_0) - f_j^{(avg)}(x_0)}{\max_x \|\nabla(f_c^{(avg)}(x) - f_j^{(avg)}(x))\|_q} \\
&= \min_{j \neq c} \frac{g_j^{(avg)}(x_0)}{\max_x \|\nabla(g_j^{(avg)}(x))\|_q}
\end{aligned}
\tag{19}
$$

For each model $f^k$, assume that the minimum of the robustness bound can be achieved with the prediction result c and j. Then the robustness bounds can be deduced to:

$$
\begin{aligned}
R^k &= \frac{g_j^k(x_0)}{\max_x \|\nabla(g_j^k(x))\|_q} \\
R^{avg} &= \frac{g_j^{(avg)}(x_0)}{\max_x \|\nabla(g_j^{(avg)}(x))\|_q},
\end{aligned}
\tag{20}
$$

where $g_j^{avg}(x) = \frac{1}{N} \sum_{k=1}^{N} g_j^k(x)$. From equation 20, we deduce two results.

First, in selecting a diverse ensemble, summing the logits from each member model smooths the average prediction $g_j^{\text{avg}}(x)$ by attenuating incorrect class probabilities. This reduces the gradient norm and the denominator in Equation 20, while amplifying the correct class probabilities—thereby increasing the margin for error and boosting the numerator. As a result, the overall average robustness improves.

Second, in the case where all models in the pool are identical, we have $R^k = R^{avg}$ for all $1 \leq k \leq N$. We claim that the following property always holds: $\exists k, 1 \leq k \leq N, R^k \leq R^{avg}$. Most importantly, this property signifies that ensembles of high diversity can improve the robustness of individual models. Therefore, we can always pair a non-robust member model with a model to obtain average robustness, which is higher than the member model. To prove this property, we use proof by contradiction. Assume that $\forall k, \ 1 \leq k \leq N, \ R^k > R^{avg}$ that is:

$$
g_j^k(x_0) \max_x \|\nabla(g_j^{(avg)}(x))\|_q > g_j^{(avg)}(x_0) \max_x \|\nabla(g_j^k(x))\|_q
\tag{21}
$$

using equation 20. Since for all the models, this inequality holds, by adding them all:

$$
\sum_{k=1}^{N} g_j^k(x_0) \max_x \|\nabla(g_j^{(avg)}(x))\|_q > \sum_{k=1}^{N} g_j^{(avg)}(x_0) \max_x \|\nabla(g_j^k(x))\|_q.
\tag{22}
$$

We can move the variables that does not depend on $k$ to the outside of the summation:

$$
(\max_x \|\nabla(g_j^{(avg)}(x))\|_q) \sum_{k=1}^{N} g_j^k(x_0) > (g_j^{(avg)}(x_0)) \sum_{k=1}^{N} \max_x \|\nabla(g_j^k(x))\|_q,
\tag{23}
$$

since $g_j^{avg}(x) = \frac{1}{N} \sum_{k=1}^{N} g_j^k(x)$ we can cancel out $g_j^{avg}$ terms to obtain:

$$
(\max_x \|\nabla(\frac{1}{N} \sum_{k=1}^{N} g_j^k(x))\|_q) > \sum_{k=1}^{N} \max_x \|\nabla(g_j^k(x))\|_q.
\tag{24}
$$

However, from the triangle inequality, we know that the term on the left must satisfy:

$$
\begin{aligned}
\max_x \|\nabla(\frac{1}{N}\sum_{k=1}^{N} g_j^k(x))\|_q &\leq \max_x \frac{1}{N}\sum_{k=1}^{N} \|\nabla(g_j^k(x))\|_q \\
&\leq \frac{1}{N}\sum_{k=1}^{N} \max_x \|\nabla(g_j^k(x))\|_q,
\end{aligned}
\tag{25}
$$

which contradicts the equation 24. Therefore, the assumption does not hold, and we show that $\exists k, 1 \leq k \leq N, R^k \leq R^{avg}$. Overall, our analysis in this section shows that a diverse ensemble team can improve the robustness of individual models in the pool.

### K.2 WHY FOCAL DIVERSITY IMPROVES PERFORMANCE?

Following Partridge & Krzanowski (1997) in the context of deep neural networks, in a system of $N$ models, $P(1)$ represents one randomly chosen model $f^i$ fails on input $x$, and $P(2)$ represents two randomly chosen models, $f^i$ and $f^j$, fail simultaneously on input.

Given that $f^i$ and $f^j$ in the pool are selected, let $X$ and $Y$ represent the random variables that $f^i$ and $f^j$ make mistakes on a randomly chosen input. Then, $P(AB)$ is the actual probability that both model fails. In the case of minimum diversity, $AB$ is an independent and equal event, and $P(1) = P(AB) = P(A) = P(B)$ since all the errors made by model $i$ are followed by $j$. In the case of maximum diversity, there is no joint between events; therefore, $A$ and $B$ are disjoint $P(2) = P(AB) = 0$. Therefore, $\frac{P(1)-P(2)}{P(2)}$ is the normalized distance from minimum diversity to maximum diversity.

Following Partridge & Krzanowski (1997), we defined the focal negative correlation score by selecting a focal model and finding its inputs where it failed and calculate as $\rho^{focal}(\mathcal{M}_i; \mathcal{E}) = 1 - \frac{P(2)}{P(1)}$ which can take 0 in minimum diversity and 1 in maximum diversity. We iterate this for every model in the ensemble set to calculate focal diversity metric, $\lambda^{focal}(\mathcal{E}) = \frac{1}{|\mathcal{E}|}\sum_{\mathcal{M}_i \in \mathcal{E}} \rho^{focal}(\mathcal{M}_i; \mathcal{E})$.

The goal of Decider Agent can be defined as:

$$
\max_{\mathcal{E} \in \mathbb{E}} \lambda^{focal}(\mathcal{E}),
\tag{26}
$$

where $\mathbb{E}$ represents universal set that contains all the combinations of models having the size of $2^N - N - 1$. By substituting, $\rho^{focal}$, we can write the equation as:

$$
\max_{\mathcal{E} \in \mathbb{E}} \frac{1}{|\mathcal{E}|} \sum_{\mathcal{M}_i \in \mathcal{E}} \rho^{focal}(\mathcal{M}_i; \mathcal{E}),
\tag{27}
$$

Since each $\rho^{focal}(\mathcal{M}_i; \mathcal{E})$ depends on the entire set $\mathcal{E}$, the objective in equation 27 is a set-level optimization problem with a set-dependent reward function. Therefore, theoretically, it is hard to show individual term maximisation due to the interaction between elements. Yet we know that the optimal $\mathcal{E}^*$ maximises the average of per $\rho^{focal}$ that depend on the whole set.

Then let $f^i$ be the focal model and $f^j$ be a randomly selected model from the optimal set $\mathcal{E}^*$, and where these models have high diversity close to maximum. Then let $P(AB) \leq \epsilon, 0 \leq \epsilon$ where $\epsilon$ is very small number. Then the covariance between events $A$ and $B$ can be shown as:

$$
\text{Cov}(A, B) = \text{E}[AB] - \text{E}[A]\text{E}[B]
\tag{28}
$$

Since we select all inputs where the focal model makes errors, we have $\text{E}[A] = 1$, and if $\text{E}[AB] \leq \varepsilon$, then it follows that:

$$
\text{Cov}(A, B) = \text{E}[AB] - \text{E}[A]\text{E}[B] \leq \varepsilon - \text{E}[B].
\tag{29}
$$

In the case of maximum diversity, $\text{E}[B] = 1$ and $\varepsilon = 0$, which yields a covariance of $-1$. This indicates that maximizing focal diversity leads to a low (or even negative) error covariance between the member models.

As we have shown in section 2, the bias-variance-covariance decomposition of an ensemble estimator can be denoted as:

$$\mathbb{E}[(\hat{f}_{\text{ens}} - y)^2] = \overline{\text{Bias}} + \frac{1}{N}\overline{\text{Var}} + (1 - \frac{1}{N})\overline{\text{Covar}}. \tag{30}$$

where the covariance term equals to:

$$\overline{\text{Covar}} = \frac{1}{N(N-1)} \sum_i \sum_{i \neq j} \text{Cov}(f^i, f^j) \tag{31}$$

Since we have shown that maximizing focal diversity leads to negative covariance between member models, the covariance term in the error decomposition decreases, thereby reducing the overall ensemble error. Consequently, as the Decider Agent moves toward maximizing focal diversity to optimize its reward, it effectively selects diverse ensemble sets that yield lower error and greater robustness.

