# OpenReview forum: "Dynamic Optimizations of LLM Ensembles with Two-Stage Reinforcement Learning Agents"
_ICLR.cc/2026/Conference — Submitted to ICLR 2026_

### Official Review · Reviewer_FG51 · 2025-10-22

**Soundness:** 3
**Presentation:** 2
**Contribution:** 3
**Rating:** 4
**Confidence:** 3

**Summary:**

This paper introduces RL-Focal, a novel two-stage Reinforcement Learning (RL) agent framework designed for robust and adaptive multi-agent systems built upon existing LLMs. Extensive evaluations on five benchmarks demonstrate that RL-Focal achieves an 8.48% performance improvement with a small ensemble compared to the best individual LLM, while also offering stronger robustness.

**Strengths:**

* Well motivated.
* Improvement on BBH seems significant.
* Methods are described in detail

**Weaknesses:**

The manuscript has substantial room for improvement, particularly in the representation and experimental design.
* The manuscript's structure seems unbalanced. Only two of the nine pages are dedicated to describing experimental results. Given the apparent lack of a theoretical contribution, the content devoted to methods and general descriptions should be significantly compressed to allow for a deeper discussion of the findings and ablations.
* The current experiments and ablations are limited. I recommend performing a detailed analysis of model selection and usage. Specifically the activation frequency or utilization ratio of each individual model within the agent system across different tasks.
* Since models like Llama 3 and Mixtral show strong inherent performance (e.g., on MMLU), are these models frequently selected for the ensemble within the proposed agent system?

To better understand the system's core capabilities, please consider the following baselines:
* Evaluate the ensemble performance of only the top two or three best-performing models on a specific task to establish a powerful baseline.
* Evaluate the baseline that combining the top $k$ outputs (e.g., $k=5$ outputs) from the best models (best of N).
* Computational cost and improvement compared to these baselines.

**Questions:**

See weaknesses.

---

> ### Author Response · Authors · 2025-11-23
>
> We are grateful to Reviewer FG51 for the insightful and constructive comments, and we greatly appreciate the reviewer's careful evaluation and effort. Below, we provide the responses to the concerns and questions of the reviewer by addressing the changes we made in the new version of the paper.
>
>
> 1) **Structure and balance of the paper:** We address the concern of the reviewer by updating the paper thoroughly with more concise and straightforward narration. The paper's new version contains three new experiments inside the main paper (see Table 2-3, and Figure 3), and four new experiments at the appendix (see Appendix B C D E). While we give the results and indicate the importance of these experiments concisely in the main paper, we address them in detail in the Appendix.
>
>
> 2) **Activation Frequency of each model** Based on the feedback of the Reviewer, in Appendix F, we analyze the Decider Agent by examining model-selection frequencies across datasets (Figures 5 and 6). The horizontal stacked bar charts in the Figures show selection densities for BBH, MuSR, GPQA, GSM8K, and MMLU (57 MMLU subsets, 24 BBH subsets, and 3 MuSR subsets). The results show four key observations. (i) High-performing models (e.g., Phi-4-14B, Qwen2.5-72B, Llama-3-70B) are not universally selected, indicating that the Decider Agent adapts its choices to dataset-specific error patterns. (ii) Mixtral-8×22B, despite ranking fourth in average accuracy, is frequently selected due to its complementary error profile. (iii) The agent reduces redundant inference and cost by selecting only a small subset of models (sometimes a single model), as observed in the Temporal Sequences and Ruin Names subsets of BBH. (iv) Overall, selection density correlates with average model quality; for example, GSM8K is dominated by its top-performing models. To further analyze the dynamic behaviour and performance of the RL-Agent, we provide a data-shift experiment in Appendix D.
>
>
> 3) **Top-k Baseline Comparison** We thank the reviewer for this experiment, and we execute it in Appendix E, and Figure 9 shows the result.  To assess the role of model diversity and ensemble fusion, we evaluate RL-Focal on the MuSR and GPQA datasets using only the top-*k* models (ranked by Table 1). Figure 9 reports RL-Focal’s accuracy (bars) and Majority Voting accuracy (lines). We make two main observations. (i) RL-Focal reaches its highest performance when the full model pool is available, demonstrating the importance of diversity: even strong models can exhibit similar error modes and may not form an effective ensemble when used alone. (ii) Majority-voting results highlight that some models introduce high-variance noise, sharply degrading the accuracy of majority voting by confusing the consensus. In contrast, RL-Focal effectively isolates the most reliable predictions and produces robust fused outputs. Moreover, in terms of computational cost, RL-Focal outperforms majority voting by pruning models and preventing redundant inference. However, as we increase the size of the candidate models in the model pool, the inference cost increases depending on the Decider Agent's selection frequency.

---

> > ### Comment · Reviewer_FG51 · 2025-11-26
> >
> > Thanks for the reply especially the extended experiments. I will raise my score to 6.

---

### Official Review · Reviewer_LM4H · 2025-10-27

**Soundness:** 2
**Presentation:** 2
**Contribution:** 3
**Rating:** 4
**Confidence:** 4

**Summary:**

This paper proposes RL-Focal, a two-stage multi-agent RL framework that dynamically routes and ensembles LLMs. The stage-1 Decider agent learns to select a small subset of models per query by optimizing both error diversity and reasoning performance with task-adaptive rewards and policies. The stage-2 Fusion agent learns to resolve conflicts and fuse the selected models outputs, and a new focal diversity metric models error correlations to improve generalization in selection and fusion.

**Strengths:**

1.	Tackles an important, practical problem: adaptive, query-wise ensembling/routing among LLMs rather than static majority voting.
2.	Ablations and sensitivity analyses help understand behavior.

**Weaknesses:**

1.	**Poor writing/formatting.**  1) Lines 50–51 contain content that should not appear in the paper; please remove or rewrite appropriately. 2) The captions/layout for Figure 3 and Figure 4 have almost no spacing, which hurts readability. Please increase the vertical spacing and ensure consistent caption styling.
2.	**Overstated novelty in the problem formulation.** The paper claims to be the first to formulate LLM ensembling as a POMDP, yet prior work (e.g., RLAE[A], DER [B]) already models LLM-ensemble reasoning as an MDP. Please clarify the substantive differences between your method and prior work.
3.	**Incomplete reporting in Table 1.** Several entries are missing (marked “–”), preventing a complete comparison across datasets. Please fill in the absent results or justify why they are unavailable.
4.	**Lack of same-setting SOTA baselines in the main table.** Table 1 compares RL-Focal primarily against base models; strong ensemble/router baselines are not included there under the same pool and evaluation protocol. For fairness, include leading SOTA methods in Table 1 (or provide a unified main results table) under identical settings.
5.	**Metric inconsistency for Qwen2.5-72B on MMLU.** The paper reports 75.01, whereas widely cited numbers are around 86.1 [C].
6.	**Minor issues.** Occasionally inconsistent capitalization (“LLama” vs “LLaMA”).


[A] RLAE: Reinforcement Learning-Assisted Ensemble for LLMs, arxiv 2025.

[B] Efficient Dynamic Ensembling for Multiple LLM Experts, IJCAI 2025.

[B] Qwen2.5 Technical Report, 2024.

**Questions:**

see Weaknesses.

---

> ### Author Response · Authors · 2025-11-23
>
> We thank Reviewer LM4H for the constructive comments and greatly appreciate the reviewer’s time and effort. Below, we provide the answers to the concerns raised by the reviewers. Based on these comments, we updated our paper and mentioned the specific locations below.
>
> 1) **Writing and Formatting update:** Thank you to the reviewer for the careful reading and analysis of our paper. We carefully rewrite the introduction and go over the other parts of the paper by making it clearer and more concise. We fixed the spacing issues to ease the readability and made consistent caption styling. And we thank the reviewer for raising these concerns.
>
>
> 2) **Novelty and problem formulation:** As the reviewer correctly noted, several prior works have formulated ensemble learning as a POMDP. However, our formulation differs in that we explicitly separate model selection (routing) and output fusion (ensembling) into two coordinated agents, with their interaction mediated by a centralized critic network. In RLAE, the agent provides weights to the next token prediction of LLMS at the span-level (fixed-length). Our method differs in terms of two aspects. First, we decouple model selection from generation. While RLAE can theoretically prune models by driving their weights to zero, in practice, the policy network often assigns small but non-zero weights, preventing effective exclusion. We instead use Bernoulli sampling based on the probabilities produced by the Decider Agent, yielding strict binary routing decisions and eliminating redundant inference. Second, inference is performed only after the ensemble pool is determined. Non-selected models never receive the input query, avoiding unnecessary computation and reducing cost. In contrast, RLAE feeds the prompt to all models and performs ensembling during generation, which is significant since LLM infrastructures charge per input and output token. As the reviewer suggested a fair comparison with the same baselines (under the same pool and evaluation protocol), we compared our approach with RLAE in our updated paper (*see Table 2 and Table 3 in the updated paper*). Table 1 compares RL-Focal with RLAE and other ensemble methods in the literature by utilizing the same model pool. While other methods struggle to improve base model performance, RL-Focal improves the performance by 1.1% in MMLU and 0.38% in GSMK8. RLAE-MAPPO shows comparable performance, yet Table 2 shows that RL-Focal reduces the cost by factors of 2x and 9x. For fairness, we have followed the same setup and utilized the same model pool.
>
> 3) **Comparison with DER** The reviewer also questioned the difference between our algorithm and DER. While both methods follow MDP formulation for ensemble learning, DER is targeted for the best trajectory (route) in a cascaded LLM ensemble pipeline via Knowledge Transfer Prompt (KTP), which is fundamentally distinct from our method, where we aim to create the best ensemble pool and combine the generated outputs. Unfortunately, the authors haven't released their codes, and we cannot perform experiments with our set of models. The GSM8k results of DER at Table 2 show that they utilized 26B parameter models and reached 34.98% accuracy with their 125M ensemble model. On the contrary, RL-Focal uses two-layer MLPs in their policies, around 175k parameters, and can reach up to 70.4% with 2b models. However, we do agree that the same model pool is required to make a fair comparison.
>
> 4) **Absent Results in Table 1** In our GMS8k experiments, we utilized the inference dataset provided by previous work [1], which includes the logits of the base models with their textual outputs for 10 forward passes. This allowed us to create probabilities assigned to the answers of the base models (see Appendix F for the details of the process). In our MMLU experiments, we utilized the HuggingFace leaderboard-old (archived) containing the MMLU results of open-sourced LLMs. For BBH, MUSR, and GPQA, we used the new version of the HuggingFace leaderboard. The new leaderboard presents only MMLU-pro results; therefore, there are no MMLU results for new models such as Phi-4-14b, Gemma-2-27b, and Qwen1.5-72b. Similarly, Qwen1.5-72B does not exist in the new open-llm-leaderboard. Unfortunately, at this moment, we do not have the computational resources to run these models.
>
> 5) **Metric Inconsistancy** We apologize for the wrong captioning of Qwen1.5-72B model as Qwen-2.5-72B. In our MMLU experiments, we used Qwen1.5-72B, whereas in BBH, MUSR, and GPQA, we used Qwen-2.5-72B. We sincerely apologize for this oversight and appreciate the reviewer’s understanding.
>
> [1] Tekin, Selim Furkan, et al. "Llm-topla: Efficient llm ensemble by maximising diversity." arXiv preprint arXiv:2410.03953 (2024).

---

> > ### Comment · Reviewer_LM4H · 2025-11-25
> >
> > Thank you for your detailed rebuttal and the updates made to the manuscript. Some of my concerns are solved. But I still have the following two issues:
> > ﻿
> > 1) Table 1 currently contains some "-", making it difficult to evaluate the advantages of the proposed method fully. Although you mention a lack of computational resources to rerun these models and instead rely primarily on benchmark results, similarly, some results could actually be directly obtained from existing technical reports (e.g., Qwen2.5 Technical Report, 2024). This point has not been adequately explained in the response as to why it was not adopted.
> > ﻿
> > 2) Table 1 should include some SOTA methods for comparison rather than just the baseline model. While you present some SOTA methods in Tables 2 and 3, you have not explained why results for these methods were not also supplemented on the remaining datasets (e.g., BBH, MUSR) and uniformly incorporated into Table 1 to present a comprehensive and fair comparative view.
> > ﻿
> > I will keep following the discussions and adjust my rating if I find it suitable.

---

> ### Author Response · Authors · 2025-11-27
>
> Thank you, Reviewer LM4H, for evaluating our response and follow-up question. We are glad to address the Reviewer's concern below.
>
> 1) In our first version of Table 1, we put the scores of models that are in the candidate ensemble set to be selected and combined by RL-Focal. Therefore, we did not want to imply that we could or couldn't improve the score of a base model that we did not use. That being said, we agree with the Reviewer, and we value the scores of the listed models to be present in the paper. Therefore, in our new version of the paper, the scores that are marked with \* are taken from the corresponding model's technical report. In addition, we want to show that the best score listed in Table 1 is reached by RL-Focal. For that reason, we obtained predictions of Qwen-2.5-72b in MMLU and GSM8k datasets using the Togetherai cloud platform, and rerun our MMLU and GSM8k experiments by including the model in the candidate model pool. We reached 93.89 in GSM8k and 83.59 in MMLU, surpassing Qwen-2.5-72b by 1.9%, and 0.91% in MMLU. The updated Table 1 is shown in the new version of the Paper.
>
> The reason for the gap (~3\%) between reported Qwen2.5-72B and our score could be based on two reasons. (1) We used the Qwen-2.5-72b-Instruct-Turbo model, which is an optimized version of a base model that is engineered for faster inference, with a trade-off in raw intelligence. (2) The difference in determining the final decision of the model. For the probabilities assigned to the choices in an MCQ, we aggregate the probabilities of the tokens creating the whole choice to compute the probability of an answer, as a method adapted by [1] [2]. After repeating the procedure for all the choices, we obtain the probability distribution over the choices, and we determine the highest probability as the final answer.
>
> 2) Unfortunately, the authors of RLAE did not share their code in public (in their EMNLP or arxiv versions, there is no link to the indicated repository). Therefore, we used the reported scores in Table 1 in their paper to compare with the RL-Focal score. And we use answers of base models to fit the RL-Focal model. If the reviewer pleases, we can merge tables 1 and 2 together, but the scores of BBH and MUSR will be represented with the "-" character. Currently, we compare our approach with RLAE using three datasets (GSM8k, MMLU, and GPQA) and the same model pool (Llama-3.1-8B-Instruct, Qwen-2-7B-Instruct, and Qwen-2.5-7B-Instruct) in Table 2.
>
> [1] Leo Gao et. al. A framework for few-shot language model evaluation, 12 2023. URL https://zenodo.org/records/10256836.
> [2] Edward Beeching et. al. Open llm leaderboard. https://huggingface.co/spaces/HuggingFaceH4/open_llm_leaderboard, 2023.

---

### Official Review · Reviewer_twgk · 2025-10-30

**Soundness:** 3
**Presentation:** 3
**Contribution:** 3
**Rating:** 6
**Confidence:** 2

**Summary:**

Introduces RL-Focal. This uses RL to route queries to the best subset of LLMs from a pool. Then another agent fuses the ensembles responses together. They also introduce a new focal diversity metric to improve pruning performance. The paper demonstrates an effective performance increase against popular benchmarks and baseline methods.

**Strengths:**

Interesting two-stage formulation separating selection (Decider) and combination (Fusion), with a multi-agent RL formulation and a centralized critic to stabilize training. Algorithms and training loops are clearly described (Algorithm 1 and 2). Furthermore, the paper attempts cost accounting and shows wall-clock/param comparisons in Appendix E (encouraging effort to quantify cost).

**Weaknesses:**

There are some similar RL ensemble approaches which limit the novelty (i.e. RLAE can in effect prune LLMs by lowering weights near zero), although they are formulated differently. The paper motivates RL via online adaptivity, but an explicit demonstration of that advantage would clarify necessity. Furthermore, training which uses two RL policies and a centralized critic adds significant computational overhead over supervised learning methods, though this is perhaps offset by the lower inference cost. It would be nice to have some further details regarding the impact of warm starting as well as experiments task distribution shifts (i.e. start with maths, end with reasoning to highlight the strengths of RL). Furthermore, I am surprised such small networks are able learn complexities of routing and combining LLM outputs. Lastly, it would be nice to see the RL training curves (performance over episodes).

**Questions:**

Do the advantages of online RL here justify the training costs of over supervised or simpler RL frameworks? What differences in approaches leads to the difference in performance between LLM-TOPLA and RL-Focal on GSM8k?
Do you have any results for performance of the difference in performance between the warm start and final RL tuned model?
Did you experiment with higher parameter counts? Are there any intuitions behind why it was effective with so few?
Are there any patterns or reasons why certain LLMs are chosen by the policy? Does this change over training time?

---

> ### Author Response · Authors · 2025-11-23
> **Response to Reviewer twgk**
>
> We want to thank the reviewer for the insightful comments and questions. We deeply appreciate their efforts that will help us to improve our work. Below, we give answers to questions and address the concerns raised by the reviewer. The reviewer can also find the locations of the updated sections, added experiments, and further discussions in our paper.
>
>
> 1) **Comparison with RLAE:** As the reviewer noted, our formulation is different, where we perform model selection (routing) and output fusion (ensemble) with two agents and establish coordination by a centralized critic net. As the reviewer claimed that RLAE can perform pruning in theory by lowering the weights to 0. First, in practice, we observe that the policy (which is modelled by NN) very frequently assigns weights that are higher than 0, even though it is very close to 0. Therefore, in our algorithm, we establish sampling via a Bernoulli distribution with the probability assigned by the Decider Agent for each model. This ensures a binary decision so that the model is selected or not, and decreases redundant inference. Secondly, we perform inference after we establish the ensemble pool; therefore, we do not even feed the input query to the model that is not selected. On the contrary, RLAE feeds the input prompt to all the models and performs an ensemble during generation. Most of the LLM Infrastructures charge by the per input and output token usage. As suggested by the reviewer, we also provide two more experiments where we compare our algorithm with RLAE (*see Table 2 and Table 3 in the updated paper*) and show that RL-focal can reach high accuracy at a lower cost. Table 1 compares RL-Focal with RLAE and other ensemble methods in the literature by utilizing the same model pool. While other methods struggle to improve base model performance, RL-Focal improves the performance by 1.1% in MMLU and 0.38% in GSMK8. RLAE-MAPPO shows comparable performance, yet Table 2 shows that RL-Focal reduces the cost by factors of 2x and 9x. For fairness, we have followed the same setup and utilized the same model pool.
>
> 2) **The online adaptability of RL-Focal:** We make an explicit demonstration of the online adaptability of RL-Focal by following the feedback of the reviewer. The resulting changes are present in Figure 3, which shows the advantage of RL-Focal's RL nature over a supervised-tuned fusion model under data distribution shift. We started from the BBH dataset and continued with GSM8k, MMLU, GPQA, MUSR, and let Algorithm-2 fit the incoming data distribution by performing model selection and fusion. RL-Focal strictly overperforms the supervised approach, showing great adaptability across datasets. We provide all the details in Appendix D about how we created the supervised model and the dataset creation. Additionally, in the updated paper, we present the frequency of selected models during inference in Figures 5 and 6 in Appendix B. The figures show how the dynamic selection Decider agent adapts to the changing data distribution shifts.
>
> 3) **The effect of Warm-start:** In Figure 8 in Appendix D, we illustrate the effect of warm-start on the performance. The warm-start makes the fusion agent more stable and increases its overall accuracy by decreasing the effect of distribution shift during data switches. We conclude that while warm-starting is not strictly necessary, it yields a consistent 2–3% improvement in accuracy by mitigating the adverse effects of distribution shift.
>
> 4) We provide the LR-Training Curves separately in Figure 7 and a discussion in Appendix C.
>
> 5) **The small sizes of the agents:** The input (the states) that is given to the models has considerably fewer dimensions (<100), which requires no need for large models. Especially, compared to vocabulary size, the input dimension is modest. However, there is still a non-linear relationship between the logits of the models; therefore, we used an NN to learn this relation. If the dimension of the solution space is high, e.g., token embeddings, we can utilize more complex models, such as in PairRanker or RLAE. Furthermore, we experienced a higher number of parameters, which resulted in overfitting with the behaviour of selecting only the same model pairs.
>
> 6) **Advantage of RL in terms of cost:** Supervised Learning Methods do not perform model pruning and require the outputs from all the models. On the other hand, RL-focal removes the redundant inferences. In terms of training, as we show in Figure 8 in Appendix D, RL-Focal without warm-start still reaches high performance with low cost.

---

> ### Author Response · Authors · 2025-12-03
> **Additional Comments**
>
> *Performance difference between LLM-TOPLA and RL-Focal on GSM8K:*
>
> There is a 0.17% performance difference between our method and LLM-TOPLA, and we provide three main reasons for this gap:
>
> (i) **Task homogeneity of GSM8K:** GSM8K measures mathematical reasoning, representing a single task genre. This is in contrast to datasets such as BBH or MMLU, which consist of a large variety of sub-tasks (e.g., math, biology, astronomy, philosophy, and others). As a result, the best-performing models on GSM8K tend to exhibit less variation compared to these more diverse benchmarks. Consequently, LLM-TOPLA, which relies on supervised fine-tuning, can fit this homogeneous setting more effectively.
>
> (ii) **Stochasticity in RL-Focal:** RL-Focal makes stochastic decisions during model selection by sampling 0/1 actions from a Bernoulli distribution. This stochasticity enables exploration and helps RL-Focal discover model combinations that generalize better to future distribution shifts. However, as a trade-off, it may temporarily include lower-performing models in the pool during exploration.
>
> (iii) **Distribution-dependent model selection:** The order of the incoming data distribution influences RL-Focal’s model selection process. For example, if easier questions dominate the early stages, the resulting model pool for later, more difficult questions may include suboptimal models, which RL-Focal only corrects after subsequent exploration. In contrast, LLM-TOPLA trains for hundreds of epochs on the training dataset and has a chance to see difficult questions throughout, allowing it to consistently learn an ensemble that performs well even on hard instances.
>
> Overall, the performance differences reflect fundamental design choices: RL-Focal is intended for dynamic, continuously shifting environments, whereas LLM-TOPLA is optimized for more static settings.

---

### Author Response · Authors · 2025-11-24
**Summary of Changes**

Dear Reviewers and ACs,

We sincerely thank all reviewers and ACs for the time, effort, and thoughtful insights provided during the review process. After considering all the comments, we conducted substantial additional experiments to address the concerns raised; the details are provided below for each response. Here, we summarize and highlight the changes in our revised manuscript compared to the first version.

1) In response to **Reviewers twgk** and **LM4H**, we compared our formulation with RLAE and stated that our difference lies in model selection and inference. Our model-pool selection uses a binary decision for each candidate model, which prevents unnecessary input and output token processing, making the system up to 9× more cost-efficient and high-performing. In the revised manuscript, we provided this comparison in Tables 2 and 3 using the same set of models, Llama-3.1-8B-Instruct, Qwen-2-7B-Instruct, and Qwen-2.5-7B-Instruct. We provided a discussion in the Experiments section for these experiments.

2) As suggested by the **Reviewers twgk** and **FG51**, we measure the activation frequency of each model by counting how many times a model is selected across multiple datasets. The resulting distribution is shown as a stacked bar chart in Figures 5 and 6. Our comments and discussion can be found in Appendix B, where we made four main observations.

3) We conducted an online-task distribution shift experiment to show the adaptability of RL-Focal, as a response to **Reviewer twgk**. We updated Figure 3 by putting the plot of performance across datasets and how RL-Focal is adapting to the change in data distribution. We started with BBH, and continue with GSM8k, MMLU, GPQA, and MUSR. As a baseline comparison, we trained a Fusion Model (MLP) using a subset of BBH and observed its performance. RL-Focal strictly outperforms the supervised model.

4) As a continuation of the adaptibility experiment and the response to the effect of warm-start by **Reviewer twgk**, we carried out online-task distribution shift without warm-start in Appendix D and Figure 8. We provide the discussion and results in Appendix D.


5) In response to **Reviewer twgk**, we put the reward curves of Decider and Fusion Agents in Figure 7 and provided a discussion in Appendix C.


6) As suggested by the **Reviewer FG51**, we measure the importance of model diversity and ensemble fusion by comparing RL-Focal performance with majority voting across top-*k* models in MUSR and GPQA. The experiment results and the discussion can be found in Appendix E.


7) As suggested by the reviewers, we made the content of the main paper more concise and made more room to include our additional experiments. The version of the paper is more balanced compared to the first version.


8) As noted by **Reviewer LM4H**, we have revised the structure of the paper by removing overly pointed statements and reorganizing the figures and captions. We also corrected the miscaptioned Qwen1.5-72B model score in Table 1.


Best regards,
Authors

---

> ### Author Response · Authors · 2025-12-03
> **Additional Changes**
>
> Dear ACs,
>
> We would like to summarize the additional changes we made after our discussion with Reviewer LM4H,
>
> 1. As requested by **Reviewer LM4H**, we filled out the scores of models labeled with '-' in Table 1 with their respective scores from their technical reports. The reason we used '-' in the first place is that those models were not in the candidate pool of the ensemble, and we haven't run inference on them. So we want to put the scores of the models we used. But we also agree with the **Reviewer LM4H**; therefore, we used the scores of those models from their technical report and labeled them with a superscript *.
>
> 2. Since we want to show that RL-Focal reaches the best result in each dataset, during the rebuttal period, we run inference on Qwen-2.5-72b-Turbo for MMLU and GSM8k datasets to include this model in the candidate ensemble pool. RL-Focal reaches 83.59% and 93.89% respectively, improving Qwen-2.5-72b-Turbo's performance relatively by 1.10% and 2.07%.
>
> 3. As recommended by the **Reviewer LM4H**, we compared our scores with RLAE using the same pool of models, [Llama-3.1-8B-Instruct, Qwen-2-7B-Instruct, Qwen-2.5-7B-Instruct] under three datasets, GSM8k, MMLU, and GPQA. The reason we cannot include MUSR and BBH is that the authors of RLAE neither provide their scores in these datasets nor release their codes publicly. When calculating the cost of RLAE, we assume that RLAE runs inference on all models in the pool, since RLAE does not perform any model pruning, while the Decider Agent first selects which combination of models will be in the pool and then runs inference. In our comparison, we concluded that RL-Focal achieves 0.76%, 0.1%, and 1.48% higher scores than RLAE, with a 2x and 9x lower inference cost.
>
> I hope we satisfy the Reviewer's concerns.
>
> We hope that our revisions adequately address the reviewer’s concerns. We respectfully believe that we have responded to all points raised, and we trust the Area Chairs to make a fair and informed decision.
>
> Best regards,
> Authors

---

### Comment · Area_Chair_kCt9 · 2025-11-25

Dear Reviewers

Thank you for your time and help for reviews.
The author-reviewer discussion due is in one week. If you have not done yet, please review the authors' rebuttal for the paper under your evaluation and engage in discussion with authors.

Thank you again.
Best,

Area Chair

---

### Meta-Review · Area_Chair_EjCR · 2026-01-07

**Summary:**

During the rebuttal, several aspects were clarified, such as the explicit online distribution-shift experiment, the warm-start ablation, and the model selection frequency. Also, the writing was improved.

But the questions on the fair comparison to the SOTA in the field remain partially unanswered (LM4H).

**Reviewer Concerns:**

Reviewer LM4H still had concerns regarding the comparison to SOTA methods.

**Reviewer Scores:**

Reviewer FG51 would raise the score. This is a borderline paper, but it leans toward rejection.

---

### Decision · Program_Chairs · 2026-01-26

Reject